# Repatterning of mammalian backbone regionalization in cetaceans

Amandine Gillet [1,2] ✉, Katrina E. Jones [1] ✉ & Stephanie E. Pierce [2] ✉

Cetacean reinvasion of the aquatic realm is an iconic ecological transition that led to drastic modifications of the mammalian body plan, especially in the axial skeleton. Relative to the vertebral column of other mammals that is subdivided into numerous anatomical regions, regional boundaries of the cetacean backbone appear obscured. Whether the traditional mammalian regions are present in cetaceans but hard to detect due to anatomical homogenization or if regions have been entirely repatterned remains unresolved. Here we combine a segmented linear regression approach with spectral clustering to quantitatively investigate the number, position, and homology of vertebral regions across 62 species from all major cetacean clades. We propose the Nested Regions hypothesis under which the cetacean backbone is composed of six homologous modules subdivided into six to nine post-cervical regions, with the degree of regionalization dependent on vertebral count and ecology. Compared to terrestrial mammals, the cetacean backbone is less regionalized in the precaudal segment but more regionalized in the caudal segment, indicating repatterning of the vertebral column associated with the transition from limb-powered to axial-driven locomotion.

The repetition of multiple homologous anatomical units within an organism, or serial homology, is widespread across the tree of life[1]. These serially homologous structures can be grouped into multiple regions, or modules, based on morphological, functional, and/or developmental features[2,3]. The level of integration (i.e., covariance) within and between each region can facilitate or constrain the acquisition of new phenotypes[4]. The axial skeleton, a serially homologous structure made of repeating vertebrae along the cranio-caudal axis, exhibits tremendous variability in regionalization patterns across vertebrates. The fish backbone is traditionally subdivided into a precaudal and caudal region, although more complex regionalization patterns have been identified in some clades[5–8]. The evolution of sturdy weight-bearing limbs in tetrapods is associated with higher vertebral regionalization including differentiation into cervical, dorsal, sacral, and caudal regions[5]. Relative to other tetrapods, the mammalian vertebral column is further regionalized, with each region characterized by distinct vertebral anatomies attained over 320 million years of mammalian evolution[9,10] (Fig. 1a).

During their evolutionary history, numerous mammalian clades independently reinvaded the aquatic environment with different degrees of aquatic adaptation and locomotor styles. The most terrestrial species, such as muskrats and shrews, rely on quadrupedal or bipedal paddling, while most aquatic species, including cetaceans and sirenians, rely solely on body oscillations[11,12]. Within cetaceans (whales, dolphins, porpoises), the shift from limb-powered to axial-powered locomotion was accompanied by a profound reorganization of the body plan[12,13]. For instance, cetaceans show an extreme reduction of the pelvis, loss of a fused sacrum, and the acquisition of a tail fin or fluke for swimming, strongly suggesting repatterning of axial skeleton regionalization[14,15]. As vertebral patterning is established early during embryonic development by the expression of collinear *Hox* genes[16,17], a repatterning of axial regionalization would suggest changes in the expression of these genes or their downstream targets.

The cetacean axial skeleton can be conveniently divided into two broad regions or segments – the precaudal and caudal segments – where

[1]Department of Earth and Environmental Sciences, University of Manchester, Manchester, UK. [2]Museum of Comparative Zoology and Department of Organismic and Evolutionary Biology, Harvard University, Cambridge, MA, USA. ✉e-mail: amandine.gillet@manchester.ac.uk; katrina.jones@manchester.ac.uk; spierce@oeb.harvard.edu

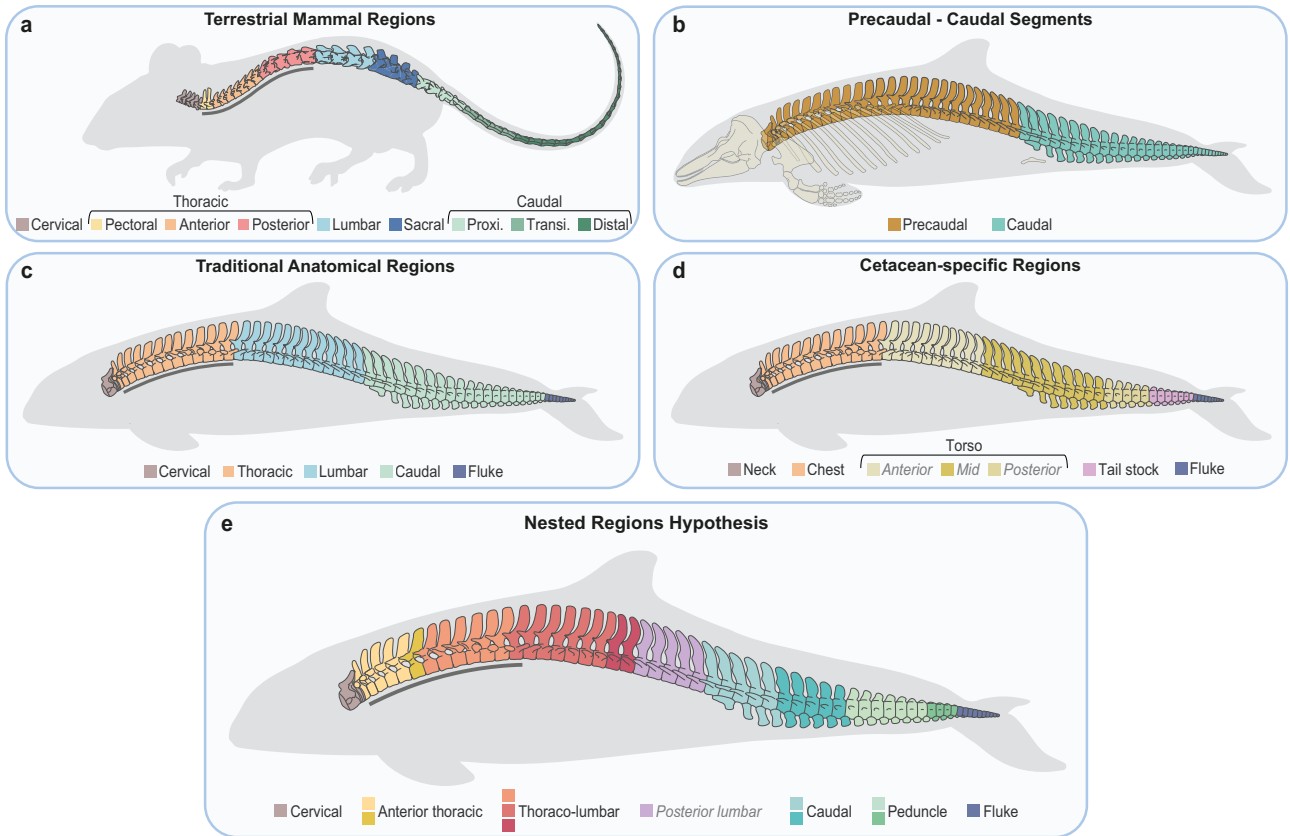

**Fig. 1 | Mammalian (a) and cetacean (b-e) backbone regionalization hypotheses.**
**a** Vertebral regions in terrestrial mammals. Presacral regions as identified by Jones et al. [9] and caudal regions as identified in New World Monkeys[39]. *Proxi.*: proximal, *Transi.*: transitional. Modified from artwork by April Neander. **b** Subdivision between precaudal (cervical, thoracic, and lumbar in **c**) and caudal (caudal and fluke in **c**) segments. **c**. Traditional mammalian anatomical regions transposed onto the cetacean backbone following Rommel[18]. Thoracic vertebrae bear ribs and caudal vertebrae bear chevrons. Fluke vertebrae are dorso-ventrally flattened. **d** Cetacean-specific regionalization pattern from Buchholtz[15,21]. The general cetacean backbone is divided into neck, chest, torso, tail stock, and fluke regions, with

the torso further subdivided into anterior, mid, and posterior in more derived oceanic dolphins. **e** Nested Regions hypothesis based on segmented linear regression and clustering analyses in this study. Vertebrae are colour-coded according to the six modules homologous across cetaceans, with the posterior lumbar module being present only in some oceanic dolphins and porpoises. Each module is composed of one to four morphological regions depending on the species, which are represented by different colour shades. For all panels, the grey bar under anterior vertebrae indicates rib-bearing vertebrae. See also Supplementary Fig. 1.

the first caudal vertebra is identified as the first vertebra bearing a chevron, or haemal arch, on the ventral-caudal aspect of the centrum[18] (although, in terrestrial mammals, the few first caudal vertebrae usually lack haemal arches[19,20]) (Fig. 1b). Beyond this, there exists disagreement in regionalization patterns. The traditional anatomical regions pattern includes subdivision into cervical, thoracic, lumbar, and caudal regions, with the addition of a distinct region in the fluke characterized by dorsoventrally flattened vertebral centra[18] (Fig. 1c). Cervical, thoracic and lumbar regions are identified based on the presence or absence of ribs and vertebrae homologous to the mammalian sacrum correspond to the posterior part of the lumbar region[20]. Alternatively, a cetacean-specific pattern dividing the backbone into neck, chest, torso, tail stock, and fluke was suggested based on variation in vertebral morphology and function[15,21] (Fig. 1d). The neck and chest correspond to the cervical and thoracic traditional regions, respectively; the torso encompasses all lumbar vertebrae and the anterior and mid caudal vertebrae; and the tail stock corresponds to laterally compressed vertebrae while vertebrae in the fluke are dorsoventrally flattened. In the most derived species, such as oceanic dolphins, the torso is further subdivided into anterior, mid, and posterior regions based on neural spine inclination[21]. Thus, there still exists a discrepancy in the exact number and homology of vertebral regions in the cetacean backbone.

Determining the number and location of vertebral regions in cetaceans is challenging due to homogenization of vertebral

morphology, with subtle cranio-caudal variation, and the broad range of vertebral counts across the clade, with 42 vertebrae in the Amazon river dolphin (*Inia geoffrensis*) and pygmy right whale (*Caperea marginata*) and up to 97 in Dall's porpoise (*Phocoenoides dalli*)[14,22]. The substantial variation in vertebral count has prevented the use of covariation methods previously used to investigate patterns of axial regionalization in other clades[2,23] as such methods are sensitive to sample size. However, new computational techniques relying on changes in linear regression slopes and maximum-likelihood to identify regionalization patterns in serially homologous structures have recently emerged[9,24] and have been successfully applied to various groups of vertebrates[25–27]. Considering regions as gradients, these methods overcome challenges associated with variable numbers of vertebrae and therefore represent a valuable tool to examine regionalization patterns in axial columns with substantial variation in vertebral count.

Here, we quantitatively investigate cetacean vertebral anatomy to assess how their backbone is patterned by identifying how many regions are present, where they are located, and how vertebral counts impact regionalization patterns. We address this by developing and applying the updated *MorphoRegions* R-package[28] to identify regionalization patterns along the post-cervical backbone of cetaceans, including species from all major clades. To homologize regions across species with vastly different vertebral counts, we further classify regions into equivalent modules using a spectral clustering approach

that does not require a priori information on the final number of modules. In addition, we quantify vertebral disparity to address whether the cetacean backbone is homogenized and how disparity and regionalization are related. Finally, since vertebral anatomy has been shown to be linked to locomotor ecology[15,21,22,29–32], we investigate whether regionalization and disparity are also related to habitat and swimming performances. Based on our results, we propose the Nested Regions hypothesis under which the cetacean backbone has been repatterned to include six morphologically homologous modules subdivided into six to nine post-cervical regions (Fig. 1e and Supplementary Fig. 1). Further, we find that vertebral count, regionalization, disparity, and ecology are interrelated suggesting that functional constraints might shape regionalization patterns. Compared to terrestrial mammals, the cetacean backbone has fewer regions in the precaudal segment but more regions in the caudal segment, including regions corresponding to the peduncle and fluke, consistent with the evolution of axial-driven locomotion.

## Results

### Vertebral regionalization and disparity

Vertebral regionalization patterns were investigated in 62 extant cetacean species based on fourteen linear and two angular measurements taken on each vertebra (Supplementary Data 1 and 2; Supplementary Fig. 2). Measurements were ordinated with a principal coordinates analysis (PCO) for each specimen separately, and axes with variance greater than 5% were retained to investigate regionalization patterns (see Methods). Our maximum-likelihood segmented linear regression approach reveals that despite loss of some anatomical regions, such as the sacrum, the cetacean backbone still possesses numerous regions (Fig. 2c and Supplementary Fig. 1). The least regionalized backbones comprise six post-cervical regions and are generally found in river dolphins, some beaked whales, and the humpback whale (*Megaptera novaeangliae*). The most regionalized backbones, found in Dall's porpoise (*P. dalli*) and several oceanic dolphins, are composed of nine post-cervical regions, representing a higher level of regionalization than previously suggested for any cetacean[21] (Supplementary Data 3). The precaudal segment of the backbone generally has fewer regions than the caudal segment (Fig. 1), with two to four regions in the precaudal and four to six in the caudal segment (Fig. 2c).

Using ancestral state reconstructions, we visualized broad evolutionary trends in vertebral regionalization and disparity (see Methods; Fig. 3). Evolution of the region score, a continuous value reflecting the level of regionalization while accounting for uncertainty in the best number of regions for each species (see Methods), shows a clear increase in regionalization level in porpoises and oceanic dolphins which also possess the highest vertebral counts[22]. The lowest regionalization scores are mostly found in river dolphins. Vertebral disparity, quantified as the average morphological distance between successive vertebrae in a common PCO morphospace including all the vertebrae from all specimens (see Methods; Fig. 2a, b), follows an opposite trend where porpoises and oceanic dolphins have the lowest disparity levels while river dolphins and beaked whales have successive vertebrae more dissimilar to each other. When considering the axial skeleton as a whole, an increase in region score is significantly associated with a decrease in vertebral disparity ($P < 0.001$, $R^2 = 0.55$) (Supplementary Fig. 3). However, when the precaudal and caudal segments are considered individually, disparity decreases with increasing regionalization only in the caudal segment (precaudal: $P = 0.476$, $R^2 = 0.50$; caudal: $P = 0.004$, $R^2 = 0.52$) (Supplementary Table 1).

### Homology of regions

Using spectral clustering analysis on PCO scores from a common morphospace of all vertebrae and specimens (see Methods), regions were classified into six homologous modules (Fig. 2a,b and

Supplementary Fig. 1). These modules are identified as anterior thoracic, thoraco-lumbar, posterior lumbar, caudal, peduncle, and fluke based on vertebral anatomy and previous descriptions of cetacean backbone regions[21]. Four of the six modules are present in all species, but the anterior thoracic module is absent in two species (*P. dalli* and *Lissodelphis borealis*) and the posterior lumbar module, which sometimes spans the traditional lumbo-caudal transition, is only found in a few oceanic dolphins and porpoises (Fig. 2c). Vertebrae in the thoraco-lumbar, posterior lumbar, and caudal modules tend to be larger with larger apophyses than vertebrae in other modules (larger PCO1 values) while the caudal and peduncle modules differ from other modules by the presence of chevrons (larger PCO2 scores). Vertebrae from the posterior lumbar and caudal modules tend to have neural spines more anteriorly inclined and smaller metapophyses (smaller PCO3 scores) (Supplementary Fig. 4).

The anterior thoracic module comprises the most anterior portion of rib-bearing vertebrae (30 to 80% of total number of rib-bearing vertebrae; see Supplementary Fig. 5a), except in most river dolphins and some beaked whales for which this module extends up to or even beyond the traditional thoraco-lumbar transition (Fig. 2c). The boundary between the anterior thoracic and thoraco-lumbar module appears to coincide with the transition from double headed (or bicipital) ribs (articulating with vertebrae via a dorsal tuberculum and a ventral capitulum) to single headed ribs (articulating only via the tuberculum) in most species (Supplementary Fig. 5e). For context, the boundary between the pectoral and anterior dorsal regions of terrestrial mammals tends to fall more anteriorly than the transition from anterior thoracic to thoraco-lumbar modules in cetaceans, suggesting that the anterior thoracic module of cetaceans is not homologous to the pectoral region of terrestrial mammals (Supplementary Fig. 5c). Conversely, the shift from double to single headed ribs in terrestrial mammals typically corresponds to the position of the diaphragmatic vertebra and the transition from the anterior dorsal to posterior dorsal region. The thoraco-lumbar module comprises the remaining rib-bearing vertebrae and most ribless lumbar vertebrae, except in species with an additional posterior lumbar module (i.e., many oceanic dolphins) and in sperm whales (*Physeter macrocephalus*, *Kogia breviceps*, and *Kogia sima*) where the caudal module also encompasses most of the lumbar vertebrae (Fig. 2c). In the caudal segment, vertebrae in the peduncle are characterised by laterally compressed centra, while vertebrae in the fluke have dorso-ventrally flattened centra compared to vertebrae in other modules (Supplementary Fig. 6).

### Impact of vertebral count on regionalization and disparity

Given the high variation in vertebral count in cetaceans, and to assess if higher vertebral counts are associated with higher regionalization levels, the effect of increasing vertebral count on regionalization was tested using phylogenetically-corrected linear regressions (see Methods). Results show that species with higher numbers of vertebrae tend to have higher region scores ($P < 0.001$, $R^2 = 0.57$) (Fig. 4c and Supplementary Table 1); even after correction for the effect of vertebral count during regionalization computation (Supplementary Note and Supplementary Fig. 7). Although vertebral counts in the precaudal and caudal segments vary similarly across species (Fig. 4a), regionalization levels are higher in the caudal segment (Fig. 4b). The regionalization level significantly increases with vertebral count in each segment, but at a higher rate in the caudal segment (precaudal: $P = 0.010$, $R^2 = 0.11$, slope = 0.038; caudal: $P < 0.001$, $R^2 = 0.42$, slope = 0.046) (Fig. 4c).

Similar to regionalization levels, the effect of vertebral count on backbone disparity was investigated using phylogenetically-informed linear regressions to determine if successive vertebrae are more similar to each other in backbones with higher vertebral count (Methods). The vertebral disparity of the entire backbone significantly

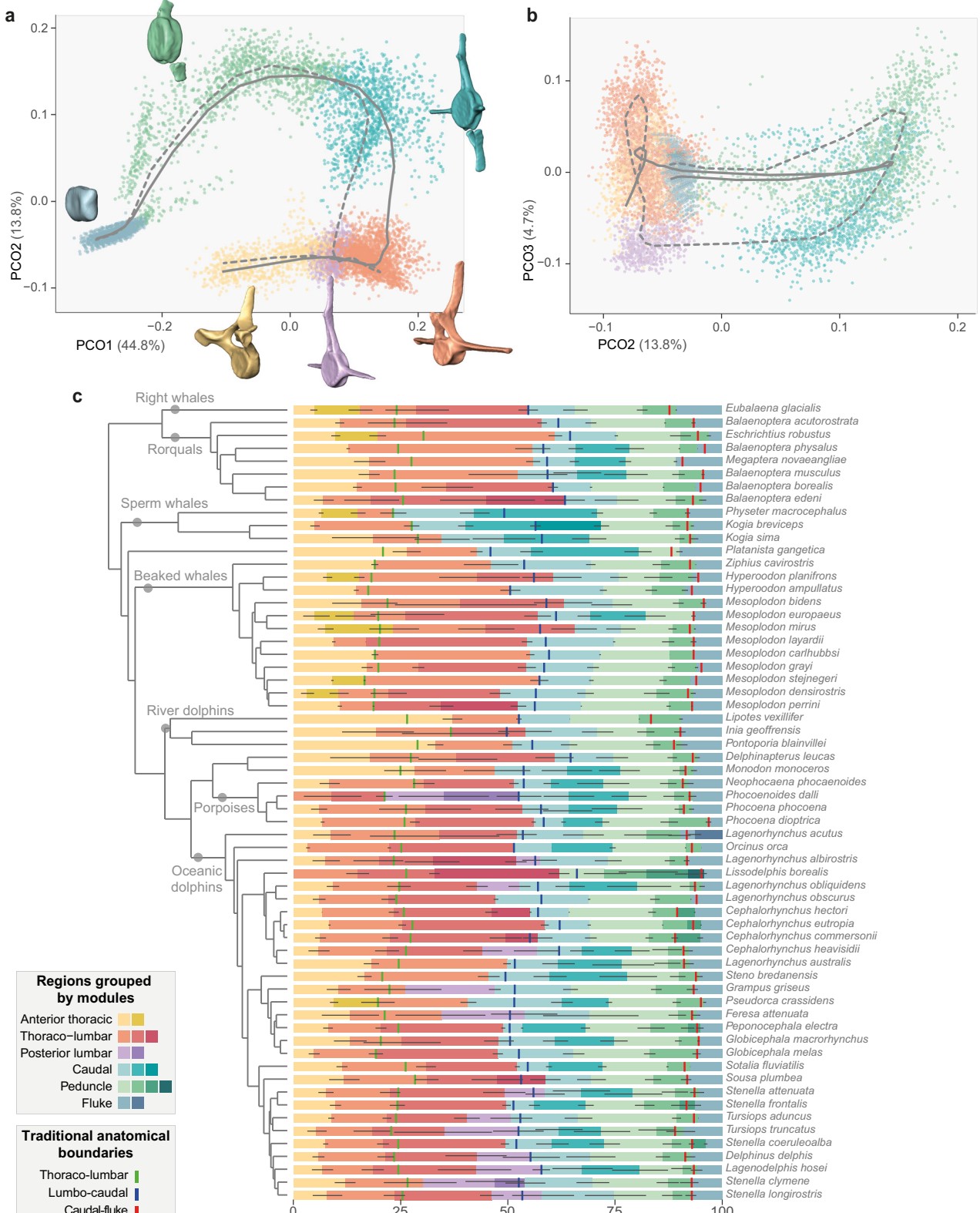

**Fig. 2 | Vertebral morphospace (a, b) and regionalization map (c).** Common morphospace of all vertebrae from all specimens derived through a principal coordinates analysis (**a:** PCOs 1 and 2, **b:** PCOs 2 and 3), with vertebrae colour-coded by the six modules recovered by spectral clusting analysis (anterior thoracic, thoraco-lumbar, posterior lumbar, caudal, peduncle, and fluke). Grey curves represent the average path from the first to the last vertebra of the backbone for specimens with (dashed line) and without (solid line) a posterior lumbar module. **c**. Vertebral maps showing regions obtained from segmented linear regressions grouped by homologous modules from spectral clustering analysis. Different modules are represented by different colours while regions are represented by different shades. Phylogeny adapted from ref. 82. See Supplementary Fig. 4 for common morphospace variable loadings, Supplementary Fig. 5 for position of anterior thoracic module boundary relative to anatomical landmarks and terrestrial mammals, and Supplementary Fig. 6 for variation in vertebral centrum shape across modules. Source data are provided as a Source Data file.

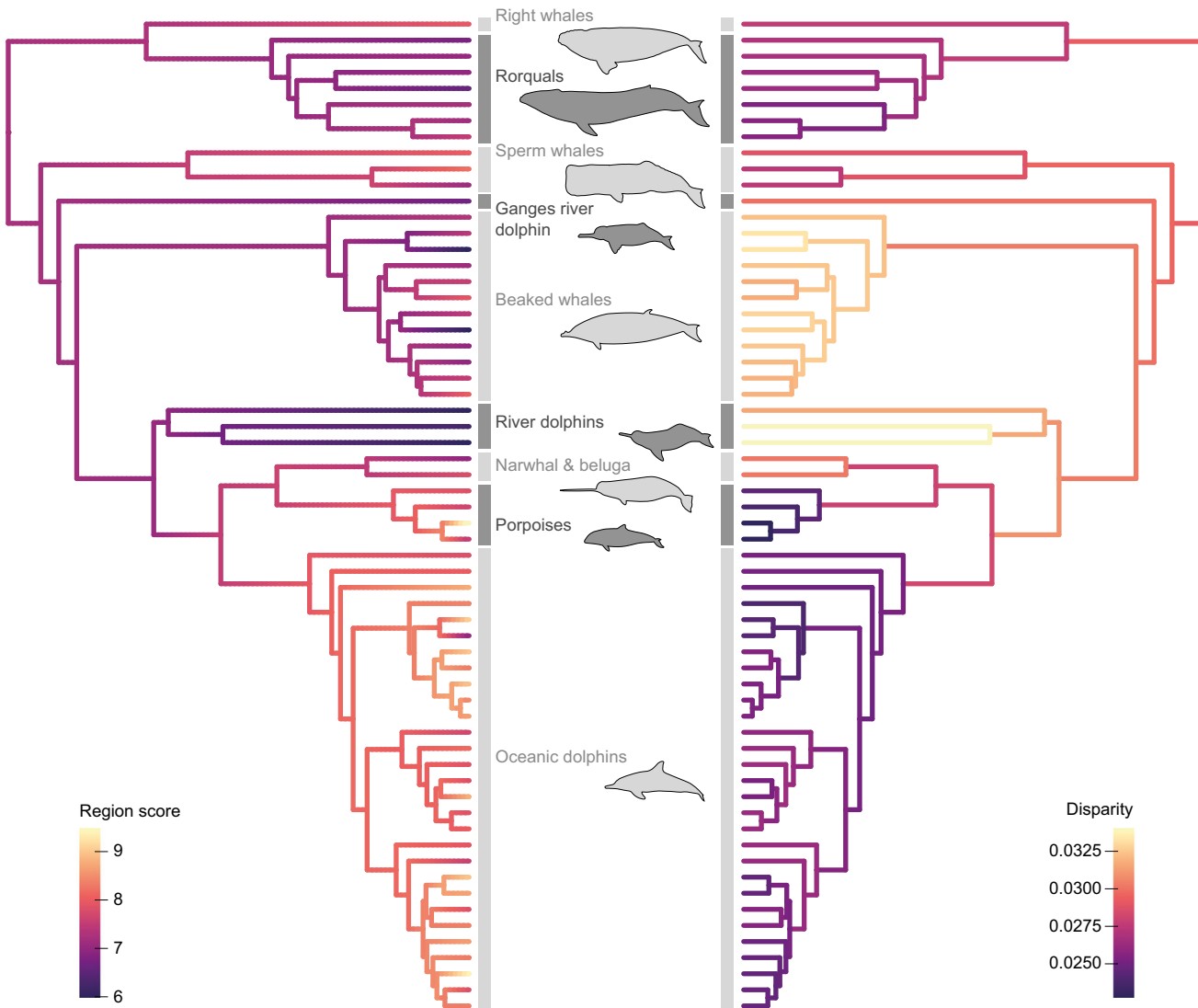

**Fig. 3 | Cetacean phylogenetic tree with mapped region score (left) and vertebral disparity (right).** Region score represents the level of regionalization (weighted average number of regions) of the backbone taking into consideration the fact that multiple models with different numbers of regions might fit the data well. Disparity corresponds to the average Euclidean distance between successive vertebrae along the backbone based on PCO scores from axes 1 to 3 of the common morphospace (see Fig. 2a, b). Oceanic dolphins and porpoises have the highest region scores but the lowest morphological disparity while river dolphins and beaked whales have high vertebral disparity but the lowest regionalization scores. Phylogeny adapted from ref. 82. See Supplementary Data 3 for detailed region score and disparity values per species. Source data are provided as a Source Data file.

decreases with increasing vertebral count ($P < 0.001$, $R^2 = 0.79$), meaning that species with higher vertebral count have adjacent vertebrae more similar to each other (Fig. 4d and Supplementary Table 1). A similar decrease in disparity is observed in both the precaudal and caudal segments (precaudal: $P < 0.001$, $R^2 = 0.58$; caudal: $P < 0.001$, $R^2 = 0.72$). The disparity in the caudal segment decreases at a higher rate but always remains higher than precaudal disparity, implying that caudal vertebrae are more dissimilar to each other than precaudal vertebrae (Fig. 4d).

**Effect of habitat and swimming kinematics**
Since vertebral count and anatomy of cetaceans vary along an inshore-offshore ecological gradient[15,22,31], the effect of ecology on vertebral regionalization and disparity was assessed to better understand if these metrics are related to locomotor ecology. Species were classified into four different environment categories (rivers, coasts, offshore, and coast-offshore mix) and phylogenetically-corrected ANOVAs were applied on the region score and vertebral disparity (Methods; Supplementary Data 3). Results highlight that habitat

has a significant effect on the region score ($P = 0.009$, $F = 4.25$), with riverine species having lower regionalization scores than other species (Fig. 5a and Supplementary Table 2). Habitat also has a significant impact on vertebral disparity ($P < 0.001$, $F = 9.19$), with riverine species having successive vertebrae more dissimilar to each other (Fig. 5b).

Because variation in vertebral morphology might reflect differences in swimming abilities[15,30], the relationship between region scores and swimming speeds was evaluated to determine if regionalization level of the backbone might be associated with swimming performances. The effect of region score on sustained swimming speed (i.e., routine swimming speed) of 34 species and burst swimming speed (i.e., high speeds that cannot be maintained for a prolonged amount of time) of 26 species was tested using phylogenetically-corrected linear regressions (Methods; Supplementary Data 3). No significant correlation was found when considering sustained swimming speeds ($P = 0.188$, $R^2 = 0.47$); however, burst swimming speeds significantly increase with higher region scores ($P = 0.010$, $R^2 = 0.68$) (Fig. 5c and Supplementary Table 3).

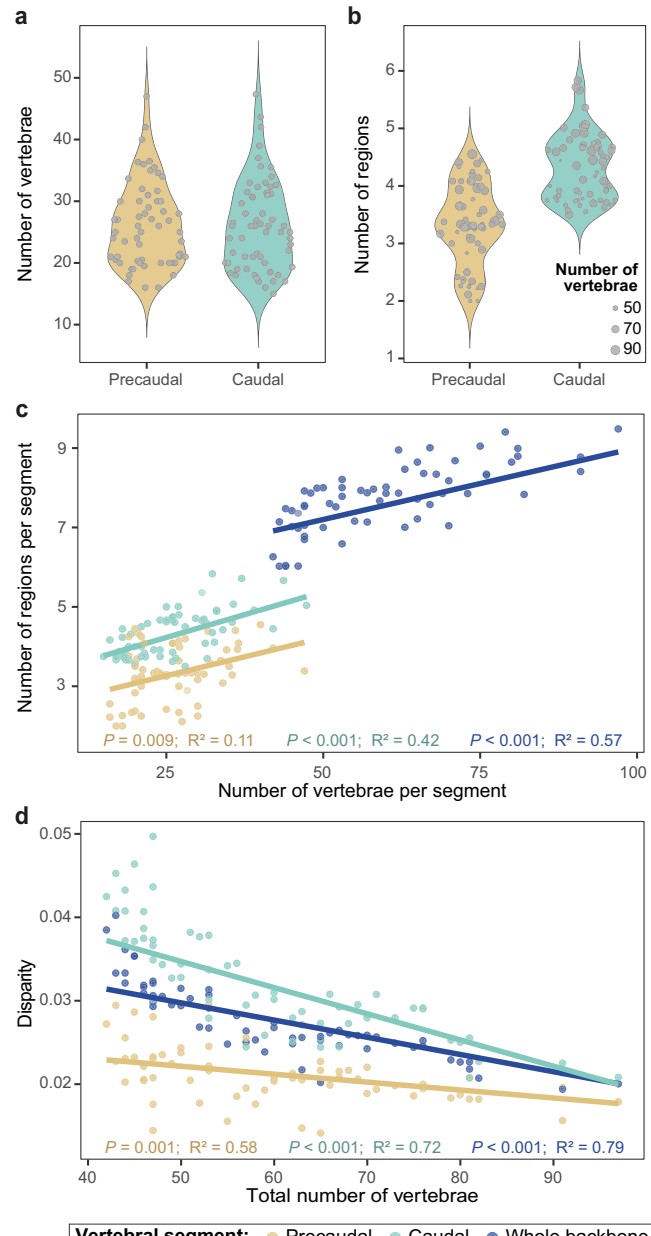

**Fig. 4 | Effect of vertebral count on regionalization and disparity.** The average number of regions is higher in the caudal segment of the backbone compared to the precaudal segment (**b**) despite similar number of vertebrae in each segment (**a**). In **b**, dot size is proportional to the number of vertebrae in the whole backbone. Phylogenetically-corrected linear regressions indicating significant increase in regionalization level (**c**) and significant decrease in disparity (**d**) with increasing vertebral count in each segment. Dots and regression lines are colour-coded by segment (whole backbone: blue, precaudal: sand, caudal: teal). See Supplementary Table 1 for detailed results of the linear regressions. Source data are provided as a Source Data file.

## Discussion

During the land-to-water transition, the cetacean body plan and underlying skeleton underwent drastic modifications associated with a streamlined body shape[33]. The loss of a well-defined sacrum, drastic reduction of the hindlimbs, acquisition of a tail fluke, substantial increase in vertebral count in some clades, and relative homogenization of vertebral morphology have made identification of anatomical vertebral regions challenging. Here, we combined segmented linear regressions and spectral clustering methods to quantify vertebral

regionalization in cetaceans and propose the Nested Regions hypothesis in which the cetacean backbone has been repatterned into a series of homologous modules that are subdivided into a number of distinct anatomical regions (Figs. 1e and 3 and Supplementary Fig. 1). Specifically, we find that the cetacean backbone is composed of six to nine post-cervical anatomical regions grouped into six homologous modules: anterior thoracic, thoraco-lumbar, posterior lumbar, caudal, peduncle, and fluke; with the posterior lumbar module being present only in some porpoises and oceanic dolphins.

Prior to our study, two conflicting regionalization patterns were proposed, primarily based on a qualitative assessment of vertebral morphology and function. The first suggested a traditional anatomical pattern in which backbone regions were homologous to terrestrial mammals – cervical, thoracic, lumbar, and caudal – but with the addition of a fluke[18] (Fig. 1c). In contrast, a cetacean-specific pattern was suggested, with the backbone subdivided into five regions – neck, chest (rib bearing vertebrae), torso (all lumbar and most of the caudal vertebrae), tail stock, and fluke – and with the torso further subdivided into anterior, mid, and posterior regions in some oceanic dolphins[15,21] (Fig. 1d). With up to six post-cervical modules subdivided into no less than six regions in all cetaceans (Figs. 1e and 2), our results recover a higher number of regions than the traditional anatomical and cetacean-specific models, which each identify four post-cervical regions in all cetaceans. Similar to the cetacean-specific pattern[21], our analyses detect a higher number of regions in oceanic dolphins and recover distinct peduncle and fluke modules, although these modules can be subdivided into further regions. Our results also differ from that pattern in several other aspects. First, the chest and torso regions from the cetacean-specific pattern are not recovered here; instead of a clear boundary between the chest (thoracic) and torso (lumbar and some caudal vertebrae), we find a module specific to most anterior thoracic (chest) vertebrae, and another module encompassing the remaining thoracic (chest) and lumbar vertebrae (torso). This boundary between the anterior thoracic and thoraco-lumbar modules coincides with the transition from double to single-headed ribs which, in terrestrial mammals, may correspond to the position of the diaphragmatic vertebra and the transition from the anterior dorsal to posterior dorsal region[9,34,35]. In addition, our analyses always recover a module boundary close to the transition between the precaudal and caudal segments, implying that the cetacean-specific torso region is composed of two modules (thoraco-lumbar and caudal) in most species, and three modules (extra posterior lumbar module) in oceanic dolphins and porpoises, with each module further subdivided into multiple regions (Figs. 1d,e and 2).

In terrestrial mammals, five presacral regions (cervical, pectoral, anterior dorsal, posterior dorsal, lumbar) have been identified using a similar approach based on segmented linear regressions[9] (Fig. 1a). Under this hypothesis, and considering the mammalian sacrum as a precaudal region on its own, the post-cervical precaudal skeleton of terrestrial mammals would count five regions (pectoral, anterior dorsal, posterior dorsal, lumbar, sacral). In comparison, we identified between two and four post-cervical regions in the precaudal segment of cetaceans (Figs. 2c and 4b), resulting in an apparent de-regionalization of the cetacean precaudal axial skeleton. Notably, analyses did not identify a distinct sacral region or module, although previous studies have suggested that vertebrae homologous to the sacrum roughly correspond to the most posterior part of the precaudal segment in modern cetaceans based on vertebral count and anatomy, pudendal nerve location, and vertebral ossification patterns[20,36,37]. While the posterior lumbar module may be considered homologous to the sacral region given its position along the backbone, this module was only identified in a few more derived species implying an unlikely scenario of de-regionalization of the sacrum in basal cetaceans followed by a secondary reacquisition in some oceanic dolphins and porpoises. The lack of a well-defined sacral region in our model

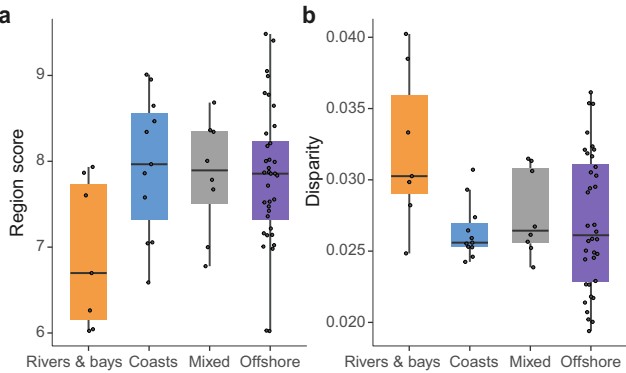

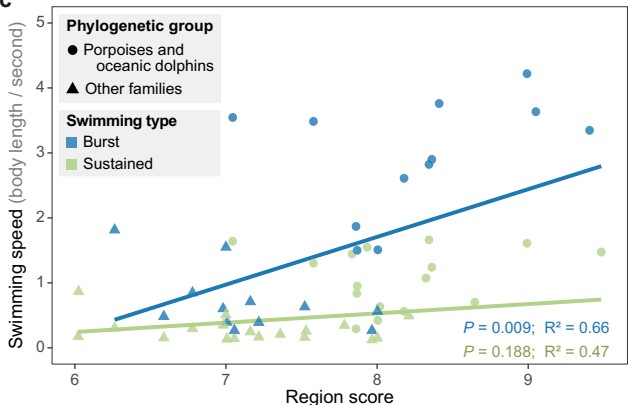

**Fig. 5 | Correlations between vertebral morphology, habitat, and swimming speed. a, b** Based on phylogenetically-corrected ANOVAs, habitat has a significant effect on region score ($P = 0.009$) and disparity ($P < 0.001$). Riverine species, which have lower vertebral counts, tend to have lower regionalization levels (**a**) but higher morphological disparity between successive vertebrae (**b**). Species (black dots) were divided in four habitat categories: rivers and bays ($n = 7$), coasts ($n = 11$), mixed ($n = 8$), and offshore ($n = 36$). For each habitat category, median (center line), first and third quartiles (box limits), and 1.5x interquartile range (whiskers). **c.** Phylogenetically-corrected linear regressions between region score and swimming speed show that species with higher vertebral regionalization levels can achieve higher speeds during burst swimming (blue line and dots), with porpoises and oceanic dolphins (circles) achieving higher speeds than other families (triangles). In contrast, there is no significant relationship between regionalization and sustained swimming speed (green line and dots). *P*-values and $R^2$ of the phylogenetically-corrected linear regressions are presented for each type of swimming speed. See also Supplementary Tables 2 and 3 for detailed results. Source data are provided as a Source Data file.

suggests that morphological differentiation of the sacral vertebrae has been overwritten with the loss of hindlimbs. Similarly, our analyses did not recover a pectoral module. In terrestrial mammals, this module is associated with the reorganization of the pectoral girdle in basal synapsids and the development of extrinsic shoulder muscles connecting the forelimb to the spine for body weight support on land[9]. Conversely, the aquatic lifestyle of cetaceans released constraints associated with body weight support and muscles connecting the forelimb to the spine (such as *m. latissimus dorsi*, *m. trapezius*, and *m. rhomboideus*) are vestigial in delphinids[38] which may have resulted in a loss of the pectoral region.

While regionalization of the caudal vertebrae in terrestrial mammals has not been quantitatively assessed, three regions (proximal, transitional, and distal) have been suggested in the tail of New World monkeys[39,40] (Fig. 1a). With three modules composed of between four and six regions, the caudal segment of the cetacean backbone is, therefore, more regionalized than a terrestrial mammal, in part due to the presence of a peduncle made of laterally compressed vertebrae

and a fluke with dorso-ventrally flattened vertebrae (Fig. 1). Subdivision of the caudal segment into multiple vertebral regions in association with the presence of caudal fins has also been proposed in fishes and other aquatic tetrapods such as ichthyosaurs and mosasaurs[7,41–43] indicating that increased caudal regionalization in fully-aquatic vertebrates may be a recurring pattern associated with the central role of the tail in axial-driven locomotion. Overall, our results show that the land-to-water transition in cetaceans did not simply result in a global increase or decrease in the number of regions. Instead, it involved a shift from a regionalized precaudal vertebral segment in terrestrial mammals to a more regionalized caudal segment in cetaceans, suggesting a decoupling of precaudal and caudal regionalization patterns[8,44,45]. Determining how and when this repatterning evolved requires further investigation of the fossil record. However, previous investigations of vertebral anatomy, and especially vertebral centra, in Protocetidae and Basilosauridae indicate that the backbone was gradually reorganized, with loss of the sacrum and acquisition of the peduncle and tail fluke[46,47]. Within basilosaurids, *Dorudon atrox* possesses dorso-ventrally flattened vertebrae indicative of the presence of a tail fluke, but lacks a distinctive peduncle, suggesting that the fluke module was acquired before the peduncle[48].

The number of vertebrae and their positional identity are defined early during embryonic development through somitogenesis and *Hox* gene expression along the body axis[17,49–51]. While the number of vertebrae is tightly linked to the oscillatory frequency of the somitogenesis clock[52,53], the repatterning of vertebral regions suggests modifications in expression patterns of *Hox* genes or their targets in cetaceans[16,17]. A few studies have highlighted positive selection in *Hoxb1*, *Hoxd1*, *Hoxb9*, *Hoxd12*[54,55] in cetaceans and while these modifications have been interpreted in the context of hindbrain and limb development, changes in *Hox* gene expression patterns are known to result in changes to vertebral identity in terrestrial mammals[17,56]. For instance, *Hox11* paralogous mutant mice lack a morphologically distinct sacrum and show evidence of sacral and anterior caudal vertebrae taking on a lumbar-like fate[57]. Furthermore, tail elongation and increased number of caudal vertebrae in mice are associated with changes in the expression patterns of *Hoxb13* and *Hoxd13* genes[58,59]. Although no positive selection has previously been identified for cetacean *Hox11* and *Hox13* paralogs, substitution rates in *Hoxc11* and *Hoxd13* appear higher than in other mammals which might reflect changes in expression patterns (see Fig. 2 in[55] and Fig. 2 in[54], respectively). However, our understanding of *Hox* gene expression and function in whales and dolphins remains greatly limited and further investigations are needed to identify potential changes to somitogenesis and axial patterning processes associated with the reorganization of their axial skeleton.

Although our Nested Regions hypothesis provides clarity on how the cetacean backbone is patterned, we still identified substantial variation in the number of regions and vertebral disparity among species (Figs. 2c and 3). These two variables are significantly correlated in the caudal segment, with disparity decreasing with an increasing number of regions, resulting in successive vertebrae being more homogeneous to each other (Supplementary Fig. 3 and Supplementary Table 1). However, the two variables are independent of each other in the precaudal segment, indicating an evolutionary decoupling between vertebral regionalization and disparity similar to what was observed in the presacral vertebral column of stem mammals[9]. Modern cetaceans also exhibit an association between vertebral count and patterns of regionalization and disparity. Even when sampling bias is corrected for, higher vertebral count is linked with a higher number of regions, especially in the caudal segment, despite the fact that somitogenesis and axial patterning are two distinct processes during embryonic development (Fig. 4 and Supplementary Table 1). An increase in vertebral count is also related to a decrease in disparity, meaning that the anatomy of successive vertebrae is more

homogeneous in species with greater numbers of vertebrae. However, irrespective of the vertebral count, the disparity is always higher in the caudal segment compared to the precaudal segment (Fig. 4d). Contrary to other vertebrate groups[60,61], higher vertebral counts are not correlated with increased body length in most modern cetaceans; instead vertebral count is negatively correlated with body size (although weakly) in porpoises and oceanic dolphins[22], which coincides with a decrease in body size along the branch leading to Delphinida (porpoises, oceanic dolphins, and river dolphins)[62]. In addition to increasing axial stiffness[15], an increase in the number of vertebrae appears to allow more gradual morphological changes and more complex cranio-caudal patterning. The different regionalization and disparity trends observed in the precaudal and caudal segments might suggest that each segment evolves under different developmental or functional constraints.

Interestingly, the highest number of vertebral regions are observed in porpoises and oceanic dolphins, which also have the highest vertebral counts (Figs. 2c and 3). Conversely, riverine species, which possess fewer vertebrae[22], tend to have less regionalized backbones with successive vertebrae more differentiated from each other potentially reflecting larger differences in biomechanical properties among successive vertebrae and/or regions in these species (Fig. 5a, b). Dolphins and porpoises with highly regionalized backbones are able to reach higher burst swimming speeds in comparison with species with lower vertebral regionalization scores (Fig. 5c). Because vertebral count, regionalization, disparity and locomotor ecology are all interconnected, it remains challenging to assess whether regionalization and disparity evolve directly under functional and ecological constraints or if increased regionalization and decreased disparity are a consequence of increased vertebral count. In cetaceans, higher vertebral counts are associated with vertebral shortening providing higher axial stability in fast-swimming offshore species[22,30,41,63]. Furthermore, changes in vertebral count may be associated with changes in the anatomy of deep axial muscles (such as fiber length and number of insertions) which could also impact axial stiffness[64], although further comparative studies of the cetacean musculoskeletal system are needed to understand how muscle attachments change with vertebral count. Nonetheless, since higher vertebral counts are also associated with an increased number of regions, a general increase in vertebral count might also provide higher functional regionalization restricting oscillatory movements to specific regions of the body. Overall, these trends in vertebral count and regionalization may result in habitat-specific locomotor styles, with oceanic species restricting oscillations to specific vertebral regions to enhance high speed swimming in open water environments and riverine species having greater mobility across a smaller number of regions to support swimming in shallow, complex environments[15,30].

In conclusion, our quantitative investigation of backbone regionalization in cetaceans allows us to propose the Nested Regions hypothesis in which the post-cervical backbone is composed of six homologous modules – anterior thoracic, thoraco-lumbar, posterior lumbar, caudal, peduncle, fluke – with the posterior lumbar module present only in some oceanic dolphins and porpoises. These modules are further subdivided into distinct anatomical regions, with a minimum of six and a maximum of nine post-cervical regions. Species with higher vertebral counts have a more regionalized backbone, especially in the caudal segment, but have successive vertebrae with a more homogeneous anatomy indicating that increased vertebral counts allow more subtle changes in cranio-caudal patterning. Riverine species tend to have less regionalized backbones while oceanic dolphins and porpoises, which are able to reach higher swimming speeds, have the highest number of regions. Compared to terrestrial mammals, the precaudal segment of cetaceans appears less regionalized, while the caudal segment is more regionalized, indicating that the cetacean backbone was repatterned during the transition from land-to-water, as

the cetacean body plan was transformed to specialize in axial-driven locomotion.

## Methods

### Vertebral shape

We collected data on vertebral count and anatomy from 139 adult specimens from 62 extant species of cetaceans (Supplementary Data 1 and 2). Whenever possible, data was collected on several specimens from the same species to account for intraspecific variability, although the number of specimens with complete vertebral columns available in collections remains limited for some species. When the number of vertebrae varied across specimens of a given species, the highest count was retained for analyses. Vertebral anatomy was quantified using fourteen linear and two angular measurements taken with digital calipers and a protractor on each post-cervical vertebra of the backbone, including chevrons in the caudal region (Supplementary Fig. 2). This totaled 7594 vertebrae and over 94,000 non-zero measurements. For specimens where all chevrons were detached from vertebrae, the vertebral position of the first chevron was identified by the presence of articular facets at the ventro-caudal face of the vertebra, and the first caudal vertebra was identified as the first vertebra bearing a chevron on the ventro-caudal face of the vertebral centrum[18] (Fig. 1b and Supplementary Fig. 1). The number of thoracic vertebrae was defined by the presence of articular facets on vertebrae and by counting the number of ribs for each specimen. The type of articulation between the rib and vertebra (double or single-headed rib) was also noted for each thoracic vertebra. Cervical vertebrae were excluded due to their variable level of fusion across species making it difficult to measure with accuracy. Since cervical vertebrae were previously identified as a distinct morphological region in terrestrial mammals[9], and the cervical region is highly constrained in mammals[65–67], we assume they also form a region in cetaceans.

Vertebral anatomy may vary serially along the column, so it is necessary to account for the loss or gain of structures when taking measurements. For example, some vertebral apophyses such as neural spines, transverse processes, or metapophyses, might be naturally extremely reduced or absent, especially in the fluke where vertebrae are simplified. Where apophyses were naturally absent on vertebrae, linear and angular measurements were treated differently to avoid artificially creating morphological changes along the backbone by coding drastically different measurements. All linear measurements for an absent apophysis were coded as zero. However, in cases where metapophyses were absent on a few successive lumbar vertebrae, their height on the vertebra (Hm; Supplementary Fig. 2) was set to the mean Hm value of the two surrounding vertebrae on which metapophyses were present since the height of metapophyses in the lumbar region is generally different from zero. Because angular measurements (antero-posterior inclination of the neural spine, Inp, and transverse process, Itp; Supplementary Fig. 2) of present apophyses were always higher than zero, angular measurement of absent apophyses were never set to zero. Where an apophysis was absent in the most caudal part of the skeleton, its angular measurement was defined as the corresponding value of the closest vertebra still bearing the apophysis. Where the apophyses were absent at a midpoint along the spine, its angular measurement was defined as the mean value of the two surrounding vertebrae bearing the apophysis. In contrast to naturally absent apophyses, in a few instances, some vertebrae were damaged resulting in broken apophyses. In such case, measurements for the damaged apophysis were interpolated by taking the mean values of the surrounding vertebrae using the *process_measurements* function from the *MorphoRegions* R-package (v.0.1.0)[28] (R v.4.0.5[68]).

Morphometric measurements were standardized prior to subsequent analyzes. For each specimen individually, variables were z-transformed using the *scale* function in R. Vertebral regionalization, clustering, and disparity analyses were all run on specimen data and

results were averaged across specimens of the same species. Number of regions and disparity were investigated for the entire backbone as well as between precaudal (thoracic and lumbar vertebrae) and caudal (all chevron-bearing vertebrae and the terminal vertebrae in the fluke) segments (Fig. 1b).

### Regionalization analysis

The number of anatomical vertebral regions along the backbone was estimated using segmented linear regression in the *MorphoRegions* R-package, an updated version of the *regions* package on GitHub[9]. This approach allows to detect changes along serially homologous structures based on variations in regression slope and intercept, hence modelling regions as gradients. Besides considerably faster computation and more versatile plotting options, the new version of the package allows users to fit more complex models (more than seven regions), choose the information criterion for model selection (AICc or BIC), define the minimum number of vertebrae required for a region, and prevent breaks between regions at specific positions. The new version also allows implementation of continuous fit models (as opposed to discontinuous), where the first point of the segment aligns with the last point of the preceding segment, modeling gradational variation without large jumps in morphology, a feature that is important when analyzing serially repeating structures with subtle variation, as is the case in cetacean backbones.

For each specimen, the previously scaled vertebral measurements were ordinated with a principal coordinates analysis (PCO) using Gower distances with the function *svdPCO* from *MorphoRegions* R-package. Only scores of PC axes with a variance greater than 5% of the total variance were used in the segmented linear regressions (usually the first two or three axes). Vertebral count was highly variable (from 33 to 90 post-cervical vertebrae in this study) and sensitivity analyses revealed a correlation between number of vertebrae and the number of regions recovered by the model (Supplementary Note and Supplementary Fig. 7). To correct for this potential bias, the segmented regressions were first calculated on a fixed and reduced number of vertebrae (33 equidistant vertebrae along the backbone) for all specimens to define the best number of regions. Subsequently, the segmented regressions were run on all post-cervical vertebrae to define the exact position of the region breaks (i.e., breakpoints) but with the number of regions fixed to that from the fixed-count analysis.

To define the optimal number of regions for each specimen, PCO scores of 33 equidistant post-cervical vertebrae from the first thoracic to the last caudal were supplied to the segmented regressions. Models ranging from one (no anatomical change along the backbone) to eleven regions (ten breakpoints along the backbone) were fitted using the function *calcregions* which does not require a priori information on the position of the breakpoints. The minimum number of vertebrae per region was set to three and successive linear segments were forced to be continuous. The best model for each given number of regions was selected by minimizing the total residual sum of squares across all PC axes with the *modelselect* function. Using the *modelsupport* function, the resulting eleven best models were then compared to identify the optimal number of regions using the Bayesian information criterion (BIC) calculated as:

$$BIC = n \ln\left(\frac{s}{n}\right) + k \ln(n) \qquad (1)$$

where $n$ is the number of data points (number of observations multiplied by the number of PC axes), $s$ is the total residual sum of squares across all PC axes used in the model, and $k$ in the number of parameters estimated. For continuous segmented regressions, the parameters to be estimated correspond to the slope of the linear segment for each region in each dimension, the intercept of the first segment in each dimension, and the position of each breakpoint. The number of

parameters, $k$, is hence calculated as follows:

$$k = rp + p + r - 1 \qquad (2)$$

where $r$ is the number of regions of the model and $p$ is the number of PC axes. The optimal number of regions for each specimen was obtained from the model with the highest BIC weight[9]. For species with multiple specimens, the mean BIC weight across specimens was computed for each number of regions and the model with the highest mean BIC weight was used to define the optimal number of regions. Since multiple models with different numbers of regions can fit the data well (i.e., have similar BIC weights), the region score, a continuous value reflecting the level of regionalization while accounting for this uncertainty, was also computed for each species. The region score ($RS$) corresponds to the sum of the number of regions in each model tested scaled by their respective BIC weight[9] and is calculated as:

$$RS = \sum_{i=1}^{R} i\, w_i \qquad (3)$$

where $i$ is the number of regions in a model with $R$ being the maximum number of regions fitted (here, eleven) and $w_i$ being the BIC weight of the model with $i$ regions. For instance, a species with BIC weights of 0.4 for the seven regions hypothesis, 0.6 for the eight regions hypothesis, and 0 for all the other tested hypotheses, would have a region score of 7.6, hence reflecting that the level of regionalization (i.e., weighted average number of regions) for this species is between seven and eight regions.

To identify the position of each breakpoint, the segmented linear regressions were fitted on all the post-cervical vertebrae of each specimen and only models with the predefined optimal number of regions for that species were conserved. The position of each breakpoint and its confidence interval were obtained by computing its weighted mean position and weighted standard deviation across the top 5% models (models with the lowest total residual sum of squares)[69]. The average position and standard deviation of each breakpoint in each species were then obtained by computing the combined mean and standard deviation of each breakpoint across specimens of the same species.

The average number of regions in the precaudal and caudal segments for each species was also computed for further statistical analyses. The number of regions in each segment was first obtained for each specimen. For regions spanning over the two segments, the proportion of vertebrae of that region falling in each segment was added to the number of regions of the corresponding segment. For instance, consider a specimen with two precaudal regions, three caudal regions, and a region made of eight vertebrae spanning over the two segments with five and three vertebrae in the precaudal and caudal segments, respectively. The number of precaudal regions for that specimen would be 2.625 – two complete regions and 62.5% of the region spanning the two segments – and the number of caudal regions would be 3.375 – three complete regions and 37.5% of the region spanning the two segments. The number of regions per segment was then averaged across specimens of the same species.

### Region homology

To identify homologous regions across species, a spectral clustering analysis was performed on the vertebral measurements using the *Spectrum* R-package (v.1.1)[70]. This method classifies observations (i.e., vertebrae) into different clusters, here defined as modules, without a priori information on the final number of clusters. Prior to analysis, scaled measurements of all specimens were ordinated in a single PCO (using Gower distances). Compared to the regionalization analysis for which individual morphospaces were computed for each specimen separately, this common morphospace comprises all vertebrae from all specimens, allowing to infer homology across species. Scores from

the first three PCs (63.4% of the total variance) were used to calculate the number of modules; this limited noise present on lower PC axes and corresponded to the maximum number of PCs conserved for regionalization analyses. Correlations between vertebral measurements and PC scores were obtained using the *cor* function to reflect the contribution of each variable to PC axes 1–3.

After the spectral clustering analysis assigned each vertebra of each specimen to a given module, module boundaries within the backbones of each species were determined by averaging the position of each boundary across specimens from the same species. Only modules identified in at least half of the specimens of a given species were conserved. Then, each region identified using segmented regressions was assigned to the module to which most of the vertebrae of the region belonged to, thus allowing us to homologize vertebrae and regions across species with vastly different vertebral counts. To interpret the homology of most anterior precaudal modules, the relative position of the boundary between the anterior thoracic and thoraco-lumbar modules (see Results) was compared to the position of the transition from double-headed to single-headed ribs in cetaceans. The anterior thoracic/thoraco-lumbar module boundary position was also compared to the region boundaries of terrestrial mammals using data from Jones et al. [9] For the caudal segment, modules boundaries were compared to patterns of vertebral centrum shape, which was quantified as the average ratio between centrum height and width (Hc / Wc).

### Vertebral disparity
The common morphospace of all specimens calculated for region homology (see above) was also used to quantify anatomical disparity along the backbone of each specimen. In this study, we defined disparity as the dissimilarity between successive vertebrae corresponding to the average distance between vertebrae. Disparity was calculated for the backbone as a whole, and for the precaudal (thoracic and lumbar) and caudal (including vertebrae from the fluke) segments separately. For each specimen, the Euclidean distance between each pair of successive vertebrae along the backbone was calculated using PCs 1 to 3. These values were then averaged for the entire backbone and for the precaudal and caudal segments to obtain a single disparity value for each segment for each specimen. The mean disparity per species for the whole backbone and for each segment was then calculated and used in subsequent analyses.

### Ecological data
To investigate the relationships between vertebral regionalization, disparity and ecology, we collected information on cetacean habitat and swimming speed from the literature. Data on habitat were collated from synthetic bibliographic works and species were divided in four habitat categories: rivers and bays, coastal, offshore, and mixed between coastal and offshore[71–73]. Data on swimming speed were also collected for 37 different species for which data were available[73–80] (Supplementary Data 3). Swimming speeds were categorized as either sustained (i.e., routine swimming speed) or burst (i.e., high speeds that cannot be maintained for a prolonged amount of time) and the mean value per species was computed for each category. Among the 37 species, data for both sustained and burst swimming speed was available for 23 species, data for sustained speed only was available for 11 species, and data for burst speed only was available for 3 species. The average body size of these 37 species were also obtained from the literature[71]. Swimming speed was then divided by body size and the size-corrected values were used for subsequent analyses.

### Statistical analyses
Broad evolutionary trends in regionalization and disparity patterns across the cetacean phylogenetic tree were visualized by estimating ancestral states of both region score and vertebral disparity using the *contMap* function from the *phytools* R-package (v.0.7-70)[81]. We further investigated the relationship between region score and disparity using a phylogenetically-corrected linear regression. Given that the number of regions in the precaudal and caudal segments of the backbone differed, the effect of the number of regions on vertebral disparity was also tested for each segment. Ancestral state reconstruction and all phylogenetically-corrected analyses were performed using the time-calibrated phylogeny from McGowen et al. [82] which was pruned to match the species included in this study using the *treedata* function from the *geiger* R-package (v.2.0.7)[83].

Since cetaceans have substantial variation in vertebral count[22], we used phylogenetically-corrected linear regressions to assess whether an increase in vertebral count correlated with (1) a higher number of regions, and (2) a higher vertebral disparity. For both traits, the effect of vertebral count was tested for the whole backbone and for each segment (precaudal and caudal). To test whether cetacean locomotor ecology impacts vertebral regionalization, the effect of habitat on region score and disparity was investigated using phylogenetically-corrected ANOVAs. Finally, we investigated the relationship between regionalization and swimming performances by fitting phylogenetically-corrected regressions between region score and each category of size-corrected swimming speed (sustained and burst) for species for which swimming speed data were available (34 species for sustained speed and 26 species for burst speed).

All linear regressions were run in a phylogenetic context using the *gls* function from the *nlme* R-package (v.3.1-152)[84] while estimating Pagel's lambda with the *corPagel* function from the *ape* R-package (v.5.4-1)[85]. Goodness of fit was assessed using the partial $R^2_{pred}$ from the *rr2* R-package (v.1.1.0) which is adapted for phylogenetic models[86,87]. ANOVAs to test the effect of habitat were computed using the *anova.gls*, *gls* and *corPagel* functions.

### Reporting summary
Further information on research design is available in the Nature Portfolio Reporting Summary linked to this article.

## Data availability
The list of specimens used in the study is available in Supplementary Data 1. Raw vertebral measurements generated in this study are available in Supplementary Data 2. Ecological categories, vertebral count, body size, and swimming speed gathered for each species are available in Supplementary Data 3. Data presented in Figs. 2 to 5 and Supplementary Figs. 3 to 7 are provided in a Source Data file. Source data are provided with this paper.

## Code availability
R code to conduct regionalization analyses is available on CRAN as part of the *MorphoRegions* package: https://CRAN.R-project.org/package=MorphoRegions/.

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

## Acknowledgements

We thank museum curators and staff members who granted access to specimens: Darrin Lunde, John Ososky, and Paula Bohaska from the Smithsonian National Museum of Natural History; Neil Duncan, Marisa Surovy, Eleanor Hoeger, and Sara Ketelsen from the American Museum of Natural History; Denise Hamerton, Jofred Opperman, and Noel Fouten from the Iziko Museums of South Africa; Greg Hofmeyr, Gill Watson, and Vanessa Isaacs from the Port Elizabeth Museum; Daniela Kalthoff, Peter Mortensen, Peter Nilsson, and Julia Stigenberg from the Swedish Museum of Natural History; Stefan Merker and Carsten Leidenroth from the Stuttgart State Museum of Natural History. We are grateful to Noah Greifer from the Harvard Institute for Quantitative Social Science for assistance in improving and formatting the new *MorphoRegions* R-package. We also thank Benjamin de Bivort at Harvard University for valuable suggestions regarding spectral clustering analyses and members of the Pierce laboratory for helpful discussions. This project was supported by the European Union's Horizon 2020 Research and Innovation Programme under the Marie Sklodowska-Curie grant agreement no. 101023931 and additional travelling funds from the University of Liège, the Belgian Fund for Scientific Research (FNRS), the Wallonia-Brussels Federation, the Odyssea asbl Luxembourg, and the European Union's Seventh Framework Programme (Synthesys grant no. SE-TAF-6278) to A.G.

## Author contributions

A.G., K.E.J, and S.E.P. conceived the project. A.G. collected and analyzed the data and interpreted results in consultation with K.E.J. and S.E.P. Figures and manuscript were drafted by A.G. alongside S.E.P., and with additional feedback by K.E.J. Code was originally written by K.E.J. and

expanded and refined by A.G. All authors edited the manuscript and approved the final version.

## Competing interests

The authors declare no competing interests.
