## [Peer Review File · Nature Communications]

Repatterning of mammalian backbone regionalization in cetaceansReviewers' Comments:

Reviewer #1:

Remarks to the Author:

This article addresses a long-standing question concerning the regionalization of the cetacean vertebral column, which deviates from the typical mammalian organization. The research is solid, drawing on an outstanding number of specimens and morphometric data. The findings are crucial to understand the homology of vertebral regions among vertebrates, using innovative statistical analyses, linking homology, evolution, swimming performance and habitat usage among cetaceans.

I will gladly support the publication of this work provided that my few comments are addressed.

1) The introduction is coherently structured; however, I disagree with the organization of the results. The opening paragraph is not clear, and the organization of figures and text need reconsideration. There is a considerable break or discontinuity at line 103, where the PCoA analysis and figures 3a-b should be introduced. These were conducted prior to determining the total number of vertebral regions. Furthermore, Figure 3 shows the PCoA analysis used in calculating the regionalization score in Figure 2, so it should be swapped or find another organization that put the PCoA before the mapped phylogenetic tree. The section discussing the “homology of regions” should be moved earlier and merged with the first paragraph, as both sections discuss the pronounced regionalization observed in river dolphins and beaked whales.

2) Figure 4 and lines 135–137, indicate that regionalization significantly increases with vertebral count per segment; however, the data is not presented. Given that figure 4 already displays disparity data per segment, the missing information should either be incorporated into the main figure or presented as a supplementary figure.

3) Supplementary Figure 1, defines segments, modules, and regions. Yet, these terms are not consistently applied in the discussion. For example, in lines 216-217, “we identified between two and four post-cervical regions in an apparent de-regionalization of the cetacean precaudal axial skeleton”, however, based on Fig. S1 it should be eight different post-cervical regions and two to four modules. Please check the consistency of terminology throughout the manuscript.

4) Moreover, I am intrigued about the names used to the different regions derived from your analysis. How were the names of the regions are “cervical”, “anterior thoracic”, “thoraco-lumbar”, and “lumbar”, chosen over cervical, thoracic, lumbar, and sacral? It is not clear in the text. There is implicit assumption of homology on the names used. So, how did you define that the sacrum is absent based on morphometrics? Or alternatively, what is the basis for not considering the thoraco-lumbar region of cetaceans not homologous to the lumbar region to other mammals?

5) Although I am not well familiarized with the cetacean fossil record, there is a good number of fossils detailing the land-to-sea transition of cetaceans from artiodactyls. There is a lack of any mentions of fossils, particularly stem cetaceans in the manuscript. How are stem cetacean

backbones regionalized, and what information is available that could account for the lack of a sacrum, if there is any evidence, as well as the changes on regionalization. Fossil stem cetaceans like *Ambulocetus* are almost complete, so including them in your analysis or at least discussing their vertebral morphological changes of different regions, could either support or not the names used for your modules, as well as adding information to the about the evolution of the cetacean spine, the lack of a sacrum, and the increased regionalization of the tail.

Reviewer #2:

Remarks to the Author:

This is a fascinating study building off of the lead author's previous excellent work in the same field. It proposes a new hypothesis for the organisation of the cetacean backbone using a proven approach that has been applied to terrestrial mammals previously.

The paper is very well written and the figures are excellent (although it is perhaps worth checking if the colour schemes all work for colour blind people). I have relatively few comments and edits, but all are in the attached pdf. They ask for clarification of a few points in the methods and suggest an area in the discussion that could be added.

Overall, once these minor comments are addressed I think this will be more than ready for publication. I look forward to seeing this study in the literature. Well done to the authors.

Dr Travis Park

Reviewer #3:

Remarks to the Author:

I am grateful for the opportunity to review the manuscript entitled “Repatterning of mammalian backbone regionalization in cetaceans”. The authors aim to resolve multiple definitions of the sections that comprise the cetacean spine through analysis of the vertebral columns of 62 species of extant whales and dolphins. They find that increased vertebral counts are associated with decreased vertebral disparity (shape differences) and increased numbers of discernible spinal subregions – particularly in the caudal (tail) segment. From the results, the manuscript proposes a new “Nested Regions” hypothesis to define regionalization of the cetacean spine. This involves six homologous modules of the post-cervical spine with multiple subdivisions (regions) that can differ across taxa.

The task attempted here is a challenging one, because cetaceans have highly variable numbers of vertebrae across species, alongside diverse habitats and locomotor modes. However, it appears that the authors have achieved what they set out to do and have created an important piece of research that is well suited to the scope of the journal. The manuscript is very well written: the aims are impactful and clearly stated, the methods are thoroughly rationalized and supported, the

results make sense and are interpreted well, and the figures are engaging and appropriate. The manuscript was a pleasure to read, and accompanied by appropriate supplementary detail. I found no major concerns to raise. I therefore only have some minor things to note.

MINOR COMMENTS

Abstract

Line 16: suggest including the number of species analyzed.

Introduction

Line 47: An “of” is missing between “expression” and “these”.

Lines 79-80: “Here, we quantitatively investigate cetacean vertebral anatomy to assess whether their backbone has been repatterned compared to terrestrial mammals and, if so, how it is patterned” – This sentence might cause some confusion as to whether terrestrial mammals were included in the study or not. Reference to terrestrial animals could probably be removed with respect to the study aims and the first two sentences of the paragraph instead could be combined.

Line 87: suggest changing “tackle the question of” to “address”, “reveal”, or something similarly less colloquial.

Line 113-115 (and methods): I must admit I found the concept of the Region Score, or more precisely, what it represents and how it relates to regionalization in general, the most challenging part of this manuscript to grasp. The methods do not appear to explicitly state what it represents as a value, but rather seem to describe it as a metric quantifying uncertainty. After looking through the methods of the cited paper (Jones et al., 2018), I think I get it – It appears to represent the overall best estimate of the number of potential post-cervical spinal regions that are present in a species when multiple hypotheses for different region counts result in similar BIC weights. I suggest rewording to better reflect this, or if I’m incorrect, rewording to what it actually describes.

Results

Line 128: Suggest changing “to” to “with”.

Discussion

Line 289: RE “more efficient swimming style”. This statement could use some finessing. A term such as “more efficient” is context dependent. Presumably, river dolphins are more efficient at swimming through unpredictable, closed-in environments, while oceanic dolphins are more efficient at generating high-power speed in open environments.

Along these lines, it might be worth also mentioning that an increase in vertebral count should come with an increase in insertions for deep spinal musculature. This is relevant in terms of stability during dynamic motion (e.g., through more rotatores muscles) and increased force production for generating greater speed in the open water.

From the results, the overall oceanic dolphin/river dolphin contrast appears to denote a graded

functional morphology in larger vertebral counts versus a more punctuated functionality in smaller counts which could be worth unpacking a little more as well.

Methods

Line 394: Suggest removing “almost”.

Line 408: “For each specimen, the number of regions in each segment was obtained by counting the number of regions in each segment.” – this sentence appears to be self-evident. Suggest rephrasing to better reflect what was intended.

Figures

Line 679: Remove both “in b)”. This part of the caption is describing part b as it is.

Figure 3 is fantastic with lots of detail that will be important for future researchers to reference. But I found part C difficult to read in the size provided in the proofs. I suggest the authors put further consideration into how to increase the size of part C. Perhaps as its own figure or redistributed and enlarged beneath parts A and B to fill a whole page in the final proofs.

The morphospaces of Figure 3 are also great. But is it possible to have a clearer description of what the PCO axes describe? For example, it appears that PCO₂ in Figure 3A is probably most influenced by the presence of the chevron? But the other axes are not so clear to me.

As a final note, it would be preferable to have the R script used for this study made available for review. In this case, most of the methods appear to be validated by previous research by the same authorship team, so this is less crucial. But it would have helped me to understand how some of the calculations were produced.

Again, well done on this nice piece of research.

Repatterning of mammalian backbone regionalization in cetaceans

We thank the Editor and Reviewers for their time and consideration of our manuscript. We have taken on all major comments and believe the new version of our manuscript is much improved.

Following the suggestion of Reviewer 1, we swapped Figs. 2 and 3 so that the morphospace and vertebral maps appear before the ancestral reconstructions of the region score and disparity, and we reorganized Fig. 2 (previously Fig. 3) to a whole-page figure so that panels are larger, according to Reviewer 3 comment. We also added three new Supplementary Figures (4-6) and expanded the “Homology of Regions” section of the Results to provide a better description of the morphological variation along the PCO axes (from Fig. 2a,b) and between modules. Finally, we expanded parts of the Discussion to add some points requested by the Reviewers (i.e., fossils, muscle insertions, rationale for module names/homology) and edited parts of the Methods for clarity (description of region score, difference between PCOs computed for regionalization and spectral clustering analyses, number of species included in the analyses of swimming speed).

Detailed answers to specific comments are presented below. Comments of the Reviewers are in italic, black font and our answers are in blue font.

Reviewer #1 :

This article addresses a long-standing question concerning the regionalization of the cetacean vertebral column, which deviates from the typical mammalian organization. The research is solid, drawing on an outstanding number of specimens and morphometric data. The findings are crucial to understand the homology of vertebral regions among vertebrates, using innovative statistical analyses, linking homology, evolution, swimming performance and habitat usage among cetaceans.

I will gladly support the publication of this work provided that my few comments are addressed.

We appreciate the support of the Reviewer for this study. We acknowledge that some parts of our manuscript lacked clarity and we edited these parts (as described below) to help resolve these issues.

1) The introduction is coherently structured; however, I disagree with the organization of the results. The opening paragraph is not clear, and the organization of figures and text need reconsideration. There is a considerable break or discontinuity at line 103, where the PCoA analysis and figures 3a-b should be introduced. These were conducted prior to determining the total number of vertebral regions. Furthermore, Figure 3 shows the PCoA analysis used in calculating the regionalization score in Figure 2, so it should be swapped or find another organization that put the PCoA before the mapped phylogenetic tree. The section discussing the “homology of regions” should be moved earlier and merged with the first paragraph, as both sections discuss the pronounced regionalization observed in river dolphins and beaked whales.

For clarity, the morphospaces presented in the original Fig. 3a,b are not the PCoAs used for the regionalization analysis, but the “all species” PCoA used to quantify disparity and homology

between regions. For the regionalization analysis, a PCoA was computed for each specimen separately; these are not presented in the text or supplement as there are way too many. We have added a sentence to the first section of Results, edited the 'Region homology' section of the Methods to clarify this (see text below), and have swapped Figs. 2 and 3 as suggested. This now makes sense due to reformatting the original Figure 3 (based on comments from Reviewer 3) to include a phylogeny.

We rearranged the Results section such that the 'Homology of regions' appears after 'Vertebral regionalization and disparity'. However, we believe this should remain as a separate section within the Results as it is based on its own, unique combination of analyses.

New text added to the Results to clarify the ordination of data for the regionalization analysis:
"[...] based on fourteen linear and two angular measurements taken on each vertebra (Supplementary Table 1; Supplementary Fig. 2). Measurements were ordinated with a principal coordinates analysis (PCO) for each specimen separately, and axes with variance greater than 5% were retained to investigate regionalization patterns (see Methods). Our maximum-likelihood segmented linear regression approach reveals that [...]"

New version of the text in the "Region homology" section of the Methods:
"Prior to analysis, scaled measurements of all specimens were ordinated in a single PCO (using Gower distances). Compared to the regionalization analysis for which individual morphospaces were computed for each specimen separately, this 'common morphospace' comprises all vertebrae from all specimens, allowing to infer homology across species. Scores from the first three PCs (63.4% of the total variance) were used to calculate the number of modules; [...]"

2) Figure 4 and lines 135–137, indicate that regionalization significantly increases with vertebral count per segment; however, the data is not presented. Given that figure 4 already displays disparity data per segment, the missing information should either be incorporated into the main figure or presented as a supplementary figure.

Figure 4c was edited to add data points and regressions for the precaudal and caudal segments.

3) Supplementary Figure 1, defines segments, modules, and regions. Yet, these terms are not consistently applied in the discussion. For example, in lines 216-217, "we identified between two and four post-cervical regions in an apparent de-regionalization of the cetacean precaudal axial skeleton", however, based on Fig. S1 it should be eight different post-cervical regions and two to four modules. Please check the consistency of terminology throughout the manuscript.

We checked the consistency of the terminology throughout the manuscript and corrected when necessary. However, our statement in lines 216-217 (now lines 245-247) is correct as our analyses recovered between 2 and 4 post-cervical, precaudal regions in all species except one (*Lagenorhynchus albirostris* which has 5 precaudal regions) as illustrated in Figs. 2c and 4b. While seven post-cervical, precaudal regions are showed in Supplementary Figure 1, this Figure is a schematic representation of the maximum number of regions identified across all species and modules. In particular, different modules can have different numbers of regions depending on the species under study, e.g., some may have more or less regions in the thoraco-lumbar module while others may have or lack a posterior lumbar region. We acknowledge that referencing Fig. S1 was misleading, and we therefore removed reference to it at the end of the referred sentence. We also edited the legend of Fig. S1 to explain that it represents the maximum number of regions per module, rather than the most common pattern.

4) Moreover, I am intrigued about the names used to the different regions derived from your analysis. How were the names of the regions are “cervical”, “anterior thoracic”, “thoraco-lumbar”, and “lumbar”, chosen over cervical, thoracic, lumbar, and sacral? It is not clear in the text. There is implicit assumption of homology on the names used. So, how did you define that the sacrum is absent based on morphometrics? Or alternatively, what is the basis for not considering the thoraco-lumbar region of cetaceans not homologous to the lumbar region to other mammals?

The thoraco-lumbar region identified here is not considered as homologous to the traditionally-defined lumbar region as, in the vast majority of species, it encompasses numerous rib-bearing vertebrae (“thoracics”) in addition to ribless vertebrae (“lumbar”). For the sacrum, our analyses do not consistently recover a well-defined region or module encompassing vertebrae that might be homologous to the sacrum (the last 40% of lumbar vertebrae according to Buchholtz (2017) *Evol. Dev.* 19, 190–204), and so we refrain from attempting to identify this region/module in our dataset. To justify our naming system, we have now expanded the Results section on region homology, as well as the Discussion, to further explain our decisions in the interpretation of the modules (see text below). An additional Supplementary Figure 4 has also been included, showing the weightings of the morphometrics variables in the common morphospace.

New text in the Results:

*“The anterior thoracic module comprises the most anterior portion of rib-bearing vertebrae (30 to 80% of total number of rib-bearing vertebrae; see Supplementary Fig. 5a), except in most river dolphins and some beaked whales for which this module extends up to or even beyond the traditional thoraco-lumbar transition (Fig. 2c). The boundary between the anterior thoracic and thoraco-lumbar module appears to coincide with the transition from double headed (or bicipital) ribs (articulating with vertebrae via a dorsal tuberculum and a ventral capitulum) to single headed ribs (articulating only via the tuberculum) in most species (Supplementary Fig. 5e). For context, the boundary between the pectoral and anterior dorsal regions of terrestrial mammals tends to fall more anteriorly than the transition from anterior thoracic to thoraco-lumbar modules in cetaceans, suggesting that the anterior thoracic module of cetaceans is not homologous to the pectoral region of terrestrial mammals (Supplementary Fig. 5c). Conversely, the shift from double to single headed ribs in terrestrial mammals typically corresponds to the position of the diaphragmatic vertebra and the transition from the anterior dorsal to posterior dorsal region. The thoraco-lumbar module comprises the remaining rib-bearing vertebrae and most ribless lumbar vertebrae, except in species with an additional posterior lumbar module (i.e., many oceanic dolphins) and in sperm whales (*Physeter macrocephalus*, *Kogia breviceps*, and *Kogia sima*) where the caudal module also encompasses most of the lumbar vertebrae (Fig. 2c). In the caudal segment, vertebrae in the peduncle are characterised by laterally compressed centra, while vertebrae in the fluke have dorso-ventrally flattened centra compared to vertebrae in other modules (Supplementary Fig. 6).”*

New text in the Discussion:

“While the posterior lumbar module may be considered homologous to the sacral region given its position along the backbone, this module was only identified in a few more derived species implying an unlikely scenario of de-regionalization of the sacrum in basal cetaceans followed by a secondary reacquisition in some oceanic dolphins and porpoises. The lack of a well-defined sacral region in our model suggests that morphological differentiation of the sacral vertebrae has been overwritten with the loss of hindlimbs. Similarly, our analyses did not recover a pectoral module. In terrestrial mammals, this module is associated with the reorganization of the pectoral girdle in basal synapsids and the development of extrinsic shoulder muscles connecting the forelimb to the spine for body weight support on land⁹. Conversely, the aquatic lifestyle of cetaceans released

constraints associated with body weight support and muscles connecting the forelimb to the spine (such as m. latissimus dorsi, m. trapezius, and m. rhomboideus) are vestigial in delphinids³⁵ which may have resulted in a loss of the pectoral region.”

*5) Although I am not well familiarized with the cetacean fossil record, there is a good number of fossils detailing the land-to-sea transition of cetaceans from artiodactyls. There is a lack of any mentions of fossils, particularly stem cetaceans in the manuscript. How are stem cetacean backbones regionalized, and what information is available that could account for the lack of a sacrum, if there is any evidence, as well as the changes on regionalization. Fossil stem cetaceans like *Ambulocetus* are almost complete, so including them in your analysis or at least discussing their vertebral morphological changes of different regions, could either support or not the names used for your modules, as well as adding information to the about the evolution of the cetacean spine, the lack of a sacrum, and the increased regionalization of the tail.*

Thank you for this important comment. While stem cetaceans are central to our understanding of how and when backbone repatterning occurred, our primary goal here was first to determine regionalization patterns in extant whales – which required extensive methodological development due to the high vertebral count found in some species. Further, the addition of fossil data comes with its own challenges, due to incompleteness of the fossil record as well as taphonomic issues, such as missing vertebrae or processes, that can have bearing on quantitative metrics. While a longer-term goal of ours is to integrate fossil data into our framework, collecting such data has also had its challenges. Due to the pandemic, most museums were closed to external visitors for many years, plus many museums had additional closures for renovations. In fact, it wasn't until a few weeks ago that we managed to get access to the Smithsonian, after waiting for the fossil marine mammal's collections to reopen. That said, we do agree that we can pay homage to the cetacean fossil record in our Discussion and have done so by adding a few sentences on broad scale changes to the backbone in stem cetaceans as previously described in the literature:

*“Determining how and when this repatterning evolved requires further investigation of the fossil record. However, previous investigations of vertebral anatomy, and especially vertebral centra, in *Protocetidae* and *Basilosauridae* indicate that the backbone was gradually reorganized, with loss of the sacrum and acquisition of the peduncle and tail fluke^{43,44}. Within basilosaurids, *Dorudon atrox* possesses dorso-ventrally flattened vertebrae indicative of the presence of a tail fluke, but lacks a distinctive peduncle, suggesting that the fluke module was acquired before the peduncle⁴⁵.”*

Reviewer #2 :

This is a fascinating study building off of the lead author's previous excellent work in the same field. It proposes a new hypothesis for the organisation of the cetacean backbone using a proven approach that has been applied to terrestrial mammals previously.

The paper is very well written and the figures are excellent (although it is perhaps worth checking if the colour schemes all work for colour blind people). I have relatively few comments and edits, but all are in the attached pdf. They ask for clarification of a few points in the methods and suggest an area in the discussion that could be added.

Overall, once these minor comments are addressed I think this will be more than ready for publication. I look forward to seeing this study in the literature. Well done to the authors.

Dr Travis Park

We are grateful to the Reviewer for the particularly positive feedback on our work. We implemented most of the few small edits (typos, grammar) and we address below all the other comments from the PDF file provided by the Reviewer.

Regarding colours used in our figures, whenever possible, colourblind friendly colour palettes were chosen and we ensured this by passing them through a dedicated website. Although some colours become harder to distinguish on Fig. 2c (Fig. 3c in the first version of the manuscript), the succession of different hues and shades of colour still allows region boundaries to be distinguished.

Comments in attached pdf:

L.107: Please add the scientific names in parentheses after the common name, as they are the one's used in the figures.

We edited the text to add scientific names after common names throughout the manuscript. For instance:

“The least regionalized backbones comprise six post-cervical regions and are generally found in river dolphins, some beaked whales, and the humpback whale (Megaptera novaeangliae). The most regionalized backbones, found in Dall’s porpoise (P. dalli) and several oceanic dolphins, are composed of nine post-cervical regions, representing a higher level of regionalization than previously suggested for any cetacean²¹ (Supplementary Table 2).”

L.172: You state in the methods that data on swimming speed was collected for 40 species, yet the total number of species here totals 60? Do some species have both types of swimming speed? Please clarify if so.

We were indeed able to obtain speed data for both types of swimming for some species but not all (as shown in Supp. Table 2). The total number of unique species for which we have swimming speed data is 37 and not 40 as previously stated in the Methods, but the sentence in the Results stating that data for sustained swimming speed was available for 34 species and burst swimming speed for 26 species is correct.

We edited that section of the Methods to clarify this:

“Data on swimming speed were also collected for 37 different species for which data were available^{69–76} (Supplementary Table 2). Swimming speeds were categorized as either sustained (i.e., routine swimming speed) or burst (i.e., high speeds that cannot be maintained for a prolonged amount of time) and the mean value per species was computed for each category. Among the 37 species, data for both sustained and burst swimming speed was available for 23 species, data for sustained speed only was available for 11 species, and data for burst speed only was available for 3 species.”

L.270: Burin et al 2023 found a dramatic decrease in long term mean body size along the branch leading to Delphinida, perhaps hinting at a negative relationship between body size and vertebral count? However, the non-platanistid river dolphins within Delphinida are an exception. Still might be worth noting.

Burin, G., Park, T., James, T.D., Slater, G.J. and Cooper, N., 2023. The dynamic adaptive landscape of cetacean body size. Current Biology, 33(9), pp.1787-1794.

In our previous work (Gillet et al. 2019, Proc. R. Soc. B), we did indeed identify a negative (but weak) correlation between vertebral count and body size in Delphinidae and Phocoenidae. We edited the sentence to mention this and add the reference to Burin, Park, et al.:

“Contrary to other vertebrate groups^{57,58}, higher vertebral counts are not correlated with increased body length in most modern cetaceans; instead, vertebral count is negatively correlated with body size (although weakly) in porpoises and oceanic dolphins²², which coincides with a decrease in body size along the branch leading to Delphinida (porpoises, oceanic dolphins, and river dolphins)⁵⁹.”

L.289: Could you discuss whether there is any anatomical evidence for the regions/modules you have found in your analyses? Do they correspond to muscle insertions etc? I know there were osteological features mentioned in the introduction, could they be referred to here as well even if to say that they don't match?

As per our response to Reviewer 1 above, we expanded the section of the Results dedicated to the homology of regions to better describe the anatomical features of vertebrae within each module, as well as the position of module boundaries in comparison to specific landmarks, especially the presence/absence of ribs and their insertions on the vertebrae (see our answer to Reviewer 1 comment #4 for details of the new text added in the Results). We also included an additional Supplementary Fig. 4 with the morphospace variable weightings.

With regards to musculature, probably due to the complexity of axial muscle anatomy in cetaceans, detailed descriptions of muscles origin and insertion on specific vertebrae remain scarce and often limited to the most common species (i.e., *Tursiops truncatus*, in Pabst (1990) in *The Bottlenose Dolphin*), preventing an interpretation of the region/module boundaries relative to muscles insertion/origin. However, we have now added a few sentences discussing the absence of the pectoral region (compared to terrestrial mammals) and its potential relationship to the reduced extrinsic shoulder muscles of cetaceans:

New text in the discussion:

“Similarly, our analyses did not recover a pectoral module. In terrestrial mammals, this module is associated with the reorganization of the pectoral girdle in basal synapsids and the development of extrinsic shoulder muscles connecting the forelimb to the spine for body weight support on land⁹. Conversely, the aquatic lifestyle of cetaceans released constraints associated with body weight support and muscles connecting the forelimb to the spine (such as m. latissimus dorsi, m. trapezius, and m. rhomboideus) are vestigial in delphinids³⁵ which may have resulted in a loss of the pectoral region.”

L.366: I would recommend running a PCO on the 33 equidistant vertebrae separately then running the segmented regressions as the two morphospaces will be different. There may not be any difference in your results in terms of the best number of regions, but it's worth checking out the alternative ordination.

When designing the study, we made the decision to run the PCO on all vertebrae and then subsample vertebrae for the regionalization analysis to limit the potential bias resulting from subsampling. Subsampling before computing the PCO implies that the variance structure in the data is independent of the vertebrae being subsampled and that the patterns in morphospace will be unaffected. Subsampling vertebrae before computing the PCO might create artificial “jumps” in morphology in the morphospace (resulting in higher numbers of regions recovered), or, on the contrary, it might result in a loss of morphological variation along the backbone

(resulting in fewer regions recovered by the analysis). Computing the PCO on all vertebrae before subsampling ensures to capture the full vertebral variation of each specimen and that the morphospace is not biased by the vertebrae that have been subsampled.

We nonetheless tested the suggestion by the Reviewer to subsample vertebrae before computing the PCO. The differences in region score between the two methods (subsampling vertebrae after computing the PCO or subsampling vertebrae before computing the PCO) are presented in the figure below, with the top panel showing the density and mean value of region scores across all species and the bottom panel detailing differences in region score for each species. In general, the region score tends to be slightly higher when subsampling vertebrae before computing the PCO but the absolute difference in region score between the two methods remains generally between ± 0.5 and only exceeds 1 for three species (*Orcinus orca*, *Hyperoodon ampullatus*, and *Kogia sima*).

L.407: “For each specimen, the number of regions in each segment was obtained by counting the number of regions in each segment.” This sentence feels a bit clumsy, could it be reworded?

We edited this sentence and following ones to streamline this part of the Methods:

“The number of regions in each segment was first obtained for each specimen. For regions spanning over the two segments, the proportion of vertebrae of that region falling in each segment was added to the number of regions of the corresponding segments. For instance, consider a specimen with two precaudal regions, three caudal regions, and a region made of eight vertebrae spanning over the two segments with five and three vertebrae in the precaudal and caudal segment, respectively. The number of precaudal regions for that specimen would be 2.625 – two complete regions and 62.5% of the region spanning the two segments – and the number of caudal regions would be 3.375 – three complete regions and 37.5% of the region spanning the two segments. The number of regions per segment was then averaged across specimens of the same species.”

L.418: “Prior to analysis, scaled measurements of all specimens were ordinated in a single PCO (using Gower distances) to create a common morphospace for all specimens. Scores from the first three PCs (63.4% of the total variance) were used to calculate the number of modules; [...]”. Is this the same PCO as used in the regionalization analysis? Perhaps easier to state that its reused here if so.

For the regionalization analysis, separate PCOs were calculated for each specimen to ensure that regionalization results are not influenced by other species included in our dataset. However, in order to identify region homology across species and compare disparity, we needed a common morphospace for all the species and so computed another ‘common PCO’ for these analyses (morphospace presented in new Fig. 2a,b – previously Fig. 3a,b). We adjusted the Results and Methods to clarify there are two different types of PCOs (see parts of text cited in our answer to Reviewer 1 comment #1).

L.422: “Module boundaries were then mapped back onto the vertebral columns for each specimen to help identify commonality across the regions recovered in the regionalization analysis.” How are they mapped back? Is it done visually in R on the 3d models or does it supply vertebra numbers of the start and end specimens? Or something else?

We edited this part of the methods to clarify how module boundaries were identified across species and how regions were assigned to each module:

“After the spectral clustering analysis assigned each vertebra of each specimen to a given module, module boundaries within the backbones of each species were determined by averaging the position of each boundary across specimens from the same species. Only modules identified in at least half of the specimens of a given species were conserved. Then, each region identified using segmented regressions was assigned to the module to which most of the vertebrae of the region belonged to, thus allowing us to homologize vertebrae and regions across species with vastly different vertebral counts.”

L.434: “For each specimen, the 3D distance between each pair of successive vertebrae along the backbone was calculated using PCs 1 to 3.” Could this not be more accurately stated as the difference in values between PCs 1-3? I don't think 3D distance strictly correct here.

We used the Euclidean distance equation to measure the distance between adjacent vertebrae in the morphospace defined by PCs 1-to 3. Hence, the disparity doesn't strictly correspond to the difference in values, and we believe that distance is the appropriate term here. We replaced '3D distance' by 'Euclidean distance' in the text:

"For each specimen, the Euclidean distance between each pair of successive vertebrae along the backbone was calculated using PCs 1 to 3."

L.459: *"Ancestral state reconstruction and all phylogenetically-corrected analyses were performed using the time-calibrated phylogeny from McGowen et al.73." Please clarify if taxa had to be pruned from the tree.*

We edited the sentence to state that the tree was pruned:

"Ancestral state reconstruction and all phylogenetically-corrected analyses were performed using the time-calibrated phylogeny from McGowen et al.⁷⁸ which was pruned to match the species included in this study using the treedata function from the geiger R-package (v.2.0.7)⁷⁹."

L.468: *"Finally, we investigated the relationship between regionalization and swimming performances by fitting phylogenetically-corrected regressions between region score and each category of size-corrected swimming speed (sustained and burst)." Were these regressions run only for the 40 species that had the swimming speed data?*

Regressions between swimming speed and vertebral regionalization were indeed calculated only for species for which swimming speed data were available. We modified the sentence to clarify this:

"Finally, we investigated the relationship between regionalization and swimming performances by fitting phylogenetically-corrected regressions between region score and each category of size-corrected swimming speed (sustained and burst) for species for which swimming speed data were available (34 species for sustained speed and 26 species for burst speed)."

Reviewer #3 :

I am grateful for the opportunity to review the manuscript entitled "Repeating of mammalian backbone regionalization in cetaceans". The authors aim to resolve multiple definitions of the sections that comprise the cetacean spine through analysis of the vertebral columns of 62 species of extant whales and dolphins. They find that increased vertebral counts are associated with decreased vertebral disparity (shape differences) and increased numbers of discernible spinal subregions – particularly in the caudal (tail) segment. From the results, the manuscript proposes a new "Nested Regions" hypothesis to define regionalization of the cetacean spine. This involves six homologous modules of the post-cervical spine with multiple subdivisions (regions) that can differ across taxa.

The task attempted here is a challenging one, because cetaceans have highly variable numbers of vertebrae across species, alongside diverse habitats and locomotor modes. However, it appears that the authors have achieved what they set out to do and have created an important piece of research that is well suited to the scope of the journal. The manuscript is very well written: the

aims are impactful and clearly stated, the methods are thoroughly rationalized and supported, the results make sense and are interpreted well, and the figures are engaging and appropriate. The manuscript was a pleasure to read, and accompanied by appropriate supplementary detail. I found no major concerns to raise. I therefore only have some minor things to note.

We thank the Reviewer for the especially positive appraisal of our work and for highlighting specific issues in the manuscript to resolve.

MINOR COMMENTS

Abstract

Line 16: suggest including the number of species analyzed.

We added the number of species included in our study in the abstract:

“Here we combine a segmented linear regression approach with spectral clustering to quantitatively investigate the number, position, and homology of vertebral regions across 62 species from all major cetacean clades.”

Introduction

Line 47: An “of” is missing between “expression” and “these”.

We added an “of” to this sentence: *“[...] a repatterning of axial regionalization would suggest changes in the expression of these genes or their downstream targets.”*

Lines 79-80: “Here, we quantitatively investigate cetacean vertebral anatomy to assess whether their backbone has been repatterned compared to terrestrial mammals and, if so, how it is patterned” – This sentence might cause some confusion as to whether terrestrial mammals were included in the study or not. Reference to terrestrial animals could probably be removed with respect to the study aims and the first two sentences of the paragraph instead could be combined.

We removed reference to terrestrial mammals and merged the two first sentences of the paragraph:

“Here, we quantitatively investigate cetacean vertebral anatomy to assess how their backbone is patterned by identifying how many regions are present, where they are located, and how vertebral counts impact regionalization patterns.”

Line 87: suggest changing “tackle the question of” to “address”, “reveal”, or something similarly less colloquial.

We replaced “tackle the question of” by “address” in this sentence:

“In addition, we quantify vertebral disparity to address whether the cetacean backbone is homogenized and how disparity and regionalization are related.”

Line 113-115 (and methods): I must admit I found the concept of the Region Score, or more precisely, what it represents and how it relates to regionalization in general, the most challenging part of this manuscript to grasp. The methods do not appear to explicitly state what it represents as a value, but rather seem to describe it as a metric quantifying uncertainty. After

looking through the methods of the cited paper (Jones et al., 2018), I think I get it – It appears to represent the overall best estimate of the number of potential post-cervical spinal regions that are present in a species when multiple hypotheses for different region counts result in similar BIC weights. I suggest rewording to better reflect this, or if I’m incorrect, rewording to what it actually describes.

We acknowledge that the concept of region score might not be straightforward based on our explanations. We edited and expanded the section of the methods to better describe the calculation of the region score and what it represents. We hope it is now easier to understand.

New text about region score:

“Since multiple models with different numbers of regions can fit the data well (i.e., have similar BIC weights), the region score, a continuous value reflecting the level of regionalization while accounting for this uncertainty, was also computed for each species. The region score (RS) corresponds to the sum of the number of regions in each model tested scaled by their respective BIC weight⁹ and is calculated as:

$$RS = \sum_{i=1}^R i w_i$$

where i is the number of regions in a model with R being the maximum number of regions fitted (here, eleven) and w_i being the BIC weight of the model with i regions. For instance, a species with BIC weights of 0.4 for the seven regions hypothesis, 0.6 for the eight regions hypothesis, and 0 for all the other tested hypotheses, would have a region score of 7.6, hence reflecting that the level of regionalization (i.e., weighted average number of regions) for this species is between seven and eight regions.”

Results

Line 128: Suggest changing “to” to “with”.

We replaced “to” by “with” in the sentence: “Given the high variation in vertebral count in cetaceans, and to assess if higher vertebral counts are associated with higher regionalization levels, the effect of increasing vertebral count on regionalization was tested using phylogenetically-corrected linear regressions (see Methods).”

Discussion

Line 289: RE “more efficient swimming style”. This statement could use some finessing. A term such as “more efficient” is context dependent. Presumably, river dolphins are more efficient at swimming through unpredictable, closed-in environments, while oceanic dolphins are more efficient at generating high-power speed in open environments.

We removed the “possibly resulting in a more efficient swimming style” part of this sentence (“Nonetheless, since higher vertebral counts are also associated with an increased number of regions, a general increase in vertebral count might also provide higher functional regionalization restricting oscillatory movements to specific regions of the body, possibly resulting in a more efficient swimming style^{15,29}.”) and added a new sentence:

“Overall, these trends in vertebral count and regionalization may result in habitat-specific locomotor styles, with oceanic species restricting oscillations to specific vertebral regions to enhance high speed swimming in open water environments and riverine species having greater mobility across a smaller number of regions to support swimming in shallow, complex environments^{15,29}.”

Along these lines, it might be worth also mentioning that an increase in vertebral count should come with an increase in insertions for deep spinal musculature. This is relevant in terms of stability during dynamic motion (e.g., through more rotatores muscles) and increased force production for generating greater speed in the open water.

While we acknowledge that changes in vertebral count most likely impacts muscular anatomy and function, our understanding of the biomechanical implications of these changes remains extremely limited. Identifying and unifying the description of axial muscles in cetaceans is complicated due to the merging and interdigitating of muscles and tendons (see Pabst, 1990, in *The Bottlenose Dolphin*, pp. 51-67, and Cozzi et al., 2016, Locomotion (Including Osteology and Myology) in *Anatomy of Dolphins: Insights Into Body Structure and Function*, pp. 33–89). Furthermore, the function of each deep axial muscle in cetaceans are hardly known due to the challenge of using experimental approaches in these species, and because their musculoskeletal system has been dramatically reorganized during the land-to-water transition.

To the best of our knowledge, no distinct *rotatores* muscles have been described in cetaceans, the major epaxial muscles being the *m. semispinalis* (muscle limited to the most anterior part of the axial skeleton), *m. longissimus* (muscle with long fibers, inserting on the dorsal tip of neural spines), and *m. multifidus* (muscle with short fibers, lying ventrally to the *m. longissimus* and inserting on vertebral metapophyses) (following the description from Pabst 1990). However, according to Cozzi et al., the *m. transversospinalis* (formed by the *m. semispinalis* and *m. multifidus*) “gives rise to smaller muscle bundles that contact adjoining vertebrae, and possibly correspond to the rotator muscles of terrestrial mammals” (Cozzi et al., 2016, p.78). While an increased vertebral count might indeed result in more numerous insertions of the *m. longissimus* as there are more numerous neural spines, this might not be true for the *m. multifidus* as, in species with higher vertebral counts, the metapophyses become extremely reduced (or completely absent) in numerous mid-posterior lumbar (and anterior caudal) vertebrae (personal observations) which might reflect a lack of insertions of the *m. multifidus* for these vertebrae.

We also investigated the snake literature as they represent another group of limbless tetrapod relying on axial undulations and also have substantial vertebral count variations. However, in this group, there seems to be an opposite pattern between the numbers of vertebrae and the overall flexibility of their body (i.e., Jayne, 1982, *J. Morphol.* 172, and Tingle et al, 2017, *J. Evol. Biol.* 30), perhaps due to lateral movements of the spine and the fact that vertebral count increase is often accompanied by body elongation, contrary to cetaceans.

For these reasons, we refrained from making any specific assumptions on the biomechanical impact but added a sentence about the non-negligible potential impact of muscles on axial stiffness as follow:

“In cetaceans, higher vertebral counts are associated with vertebral shortening providing higher axial stability in fast-swimming offshore species^{22,29,38,60}. Furthermore, changes in vertebral count may be associated with changes in the anatomy of deep axial muscles (such as fiber length and number of insertions) which could also impact axial stiffness⁶¹, although further comparative studies of the cetacean musculoskeletal system are needed to understand how muscle attachments change with vertebral count.”

From the results, the overall oceanic dolphin/river dolphin contrast appears to denote a graded functional morphology in larger vertebral counts versus a more punctuated functionality in smaller counts which could be worth unpacking a little more as well.

We slightly expanded a part of the Discussion to mention that higher morphological disparity in species with lower vertebral counts might also reflect higher functional disparity. However, it's difficult to further develop the functional impact of changes in vertebral disparity due to the lack of comparative studies on the biomechanics of the spine across cetaceans with different vertebral anatomy (vertebral count and shape).

Newly expanded sentence:

“Conversely, riverine species, which possess fewer vertebrae²², tend to have less regionalized backbones with successive vertebrae more differentiated from each other potentially reflecting larger differences in biomechanical properties among successive vertebrae and/or regions in these species (Fig. 5a,b).”

Methods

Line 394: Suggest removing “almost”.

We removed “almost” (which was in front of “similar BIC”) from that sentence: *“Since multiple models can have similar BIC weights, a continuous value accounting for this uncertainty, the region score, was also computed for each species.”*

Line 408: *“For each specimen, the number of regions in each segment was obtained by counting the number of regions in each segment.”* – this sentence appears to be self-evident. Suggest rephrasing to better reflect what was intended.

We edited this sentence and the following one to streamline this part of the Methods:

“The number of regions in each segment was first obtained for each specimen. For regions spanning over the two segments, the proportion of vertebrae of that region falling in each segment was added to the number of regions of the corresponding segment. For instance, consider a specimen with two precaudal regions, three caudal regions, and a region made of eight vertebrae spanning over the two segments with five and three vertebrae in the precaudal and caudal segment, respectively. The number of precaudal regions for that specimen would be 2.625 – two complete regions and 62.5% of the region spanning the two segments – and the number of caudal regions would be 3.375 – three complete regions and 37.5% of the region spanning the two segments. The number of regions per segment was then averaged across specimens of the same species.”

Figures

Line 679: Remove both “in b)”. This part of the caption is describing part b as it is.

Thank you for pointing out this error. We were aiming to reference panel c, so we replaced “in b)” by “in c)” (legend of Fig.2, previously Fig.3 in the first version of the manuscript).

Figure 3 is fantastic with lots of detail that will be important for future researchers to reference. But I found part C difficult to read in the size provided in the proofs. I suggest the authors put further consideration into how to increase the size of part C. Perhaps as its own figure or redistributed and enlarged beneath parts A and B to fill a whole page in the final proofs.

We appreciate the positive feedback on this Figure. We followed the recommendation and edited the figure as a whole-page to be able to enlarge panel c. We hope this makes the figure easier to read. Note that, based on Reviewer 1's suggestion, we swapped Figs. 2 and 3, so that

the figure referenced in this comment (presenting the morphospaces and vertebral maps), is now Fig. 2 in our new version of the manuscript.

The morphospaces of Figure 3 are also great. But is it possible to have a clearer description of what the PCO axes describe? For example, it appears that PCO2 in Figure 3A is probably most influenced by the presence of the chevron? But the other axes are not so clear to me.

We acknowledge that the description of PCO axes was not very detailed. We have added Supplementary Fig. 4 presenting correlations between vertebral measurements and PCO axes and we also provide a brief description of this in the “Homology of regions” section of the results:

“Vertebrae in the thoraco-lumbar, posterior lumbar, and caudal modules tend to be larger with larger apophyses than vertebrae in other modules (larger PCO1 values) while the caudal and peduncle modules differ from other modules by the presence of chevrons (larger PCO2 scores). Vertebrae from the posterior lumbar and caudal modules tend to have neural spines more anteriorly inclined and smaller metapophyses (smaller PCO3 scores) (Supplementary Fig. 4).”

As a final note, it would be preferable to have the R script used for this study made available for review. In this case, most of the methods appear to be validated by previous research by the same authorship team, so this is less crucial. But it would have helped me to understand how some of the calculations were produced.

We appreciate this suggestion and we will make the *MorphoRegions* R-package available upon publication. A link to the unpublished version of the package is available in the Reporting Summary (under the “Software and Code” section) submitted along our manuscript. We are happy to share this link via the Editor if the Reporting Summary wasn’t made available to Reviewers.

Again, well done on this nice piece of research.

Reviewers' Comments:

Reviewer #1:

Remarks to the Author:

Dear Authors,

You have satisfactorily addressed all my comments on the manuscript, and therefore I will gladly support this amazing work for publication.

Reviewer #2:

Remarks to the Author:

The authors have more than adequately addressed the reviewers concerns in their revised version of this manuscript. I have no further suggestions to make and recommend it for publication. Well done to the authors, this will be an excellent addition to the literature.

Reviewer #3:

Remarks to the Author:

The authors have made an exemplary effort with their responses and manuscript edits as needed. I have no further comments.

Repatterning of mammalian backbone regionalization in cetaceans

We thank the Editor and Reviewers for taking the time to review our manuscript. We are extremely grateful for the positive feedback received.

Below, we addressed the remaining comment from Reviewer #1 regarding our code availability. Comments of the Reviewers are in italic, black font and our answers are in blue font.

Reviewer #1 :

Dear Authors,

You have satisfactorily addressed all my comments on the manuscript, and therefore I will gladly support this amazing work for publication.

I did not see a README file so I was not sure on how to run the code. I am not a programmer myself so I would rely on a more detailed instruction on how to run the code.

We thank the Reviewer for their support. Regarding the instructions to run the code, we have now added a README file briefly describing basic steps to run the code, as well as a more detailed vignette presenting all the functionalities of the package. These can be found on the package website: <https://aagillet.github.io/MorphoRegions/>

Reviewer #2 :

The authors have more than adequately addressed the reviewers concerns in their revised version of this manuscript. I have no further suggestions to make and recommend it for publication. Well done to the authors, this will be an excellent addition to the literature.

We are grateful to the Reviewer for their feedback and appreciation of our work.

Reviewer #3 :

The authors have made an exemplary effort with their responses and manuscript edits as needed. I have no further comments.

We thank the Reviewer for acknowledging our efforts to address previous comments.